# Discovering Important Experts for Mixture-of-Experts Models Pruning Through a Theoretical Perspective

**Weizhong Huang**[1,2]    **Yuxin Zhang**[1]    **Xiawu Zheng**[1,2,3]    **Fei Chao**[1]
**Rongrong Ji**[1,2]    **Liujuan Cao**[1,2*]

[1]Key Laboratory of Multimedia Trusted Perception and Efficient Computing,
Ministry of Education of China, Xiamen University, 361005, P.R. China.
[2]Institute of Artificial Intelligence, Xiamen University.
[3] Peng Cheng Laboratory, Shenzhen, China.

## Abstract

Mixture-of-Experts (MoE) architectures enable efficient scaling of large language models but face prohibitive memory demands due to massive parameterization. Existing pruning methods rely on heuristic metrics or impractical enumeration of expert subsets, leading to suboptimal performance or scalability. In this paper, we propose Shapley-MoE, an efficient pruning method for MoE models inspired by cooperative game theory. By quantifying each expert's contribution via Shapley value, our method identifies important experts without exhaustive combination evaluations. To overcome the NP-hard complexity of exact Shapley computation, we introduce a Monte Carlo sampling strategy for efficient approximation that reduces complexity to quadratic time. However, vanilla Monte Carlo sampling still faces issues of insufficient estimation accuracy and low sampling efficiency. To address these issues, we further propose two novel methods to improve sampling accuracy and efficiency: (1) Early Truncation, which early terminates unstable sampling steps caused by overly small expert subsets, and (2) Router-Guided Importance Sampling, which prioritize sampling important expert subsets using gating activation probabilities. Both theoretical and experimental analyses show that both methods can accelerate Shapley value estimation and improve accuracy. Extensive empirical evaluations demonstrate that our pruned MoE models outperform existing expert pruning methods. Notably, when applied to the Qwen2-57B-A14B model, our method reduces the number of experts by 25% with only a 0.92 increase in perplexity and over 96.4% of the average zero-shot accuracy is maintained.

## 1 Introduction

Mixture-of-Experts (MoE) [40, 16, 79] architectures have emerged as a popular architecture for large language models (LLMs) [67, 13, 74], enabling efficient scaling and superior performance on complex tasks [22, 5, 36]. However, its sparse activation paradigm introduces massive parameterization, posing prohibitive memory demands. To address the above issues, various MoE model compression strategies have been developed, including pruning [56, 6, 24, 10], quantization [17, 30, 41], and knowledge distillation [37, 72, 65].

Among these methods, network pruning [51, 75, 66, 49, 33, 31, 70, 32, 52] can remove less important parameters from MoE models, allowing reducing the memory footprint and computational complexity of MoE models without significantly sacrificing performance. Recent studies have shown that there is significant redundancy among experts in MoE models [9, 49], and MoE models can still maintain

---

*Corresponding author: caoliujuan@xmu.edu.cn

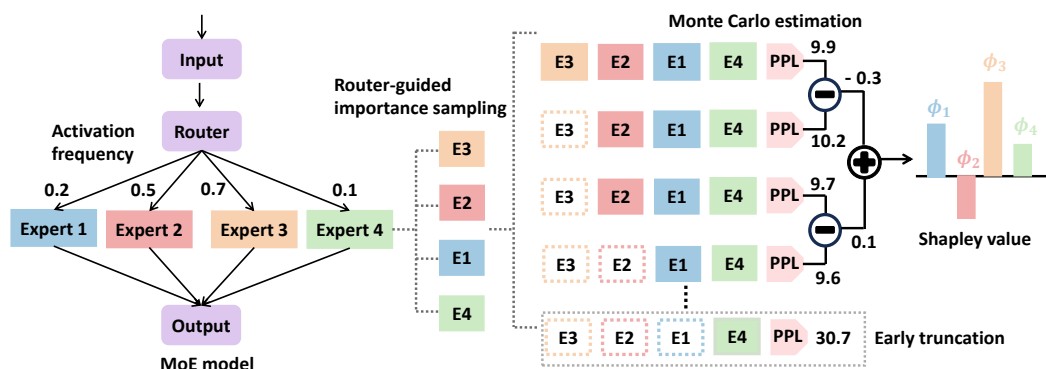

Figure 1: Shapley-MoE prunes experts by calculating their Shapley values via router-guided Monte Carlo sampling. It analyzes expert activation frequencies to prioritize important experts during sampling, iteratively removes experts to compute their marginal contributions, and applies early truncation for insignificant performance. Experts are then pruned based on their ranked Shapley values.

superior performance after redundant experts are removed. Therefore, many expert pruning methods have been proposed to remove unimportant experts from MoE models. These expert pruning methods can mainly be divided into the following two categories:

**Metric based methods.** These methods determine the importance of experts in MoE models based on manually designed metrics. For example, SEER-MoE [56] proposes pruning MoE models based on the activation frequency of experts, removing those with lower activation frequencies. He et al. [24] proposed to calculate the averaged routing score of each expert, then retain the expert with the highest score and remove the rest. Although these metric-based methods have demonstrated their effectiveness through experiments, extensive validation is required to ensure that the manually designed metrics are indeed effective. More importantly, most of these heuristic methods lack theoretical analysis, which can easily lead to suboptimal performance.

**Enumeration based methods.** These methods determine the pruned MoE model by enumerating retained expert subsets. For example, NAEE [49] enumerates all possible combinations of retained experts and selects the combination that minimizes the Frobenius norm of the output difference before and after pruning. CD-MoE [6] uses a greedy search method to obtain the pruned model. It starts with an empty subset of retained experts and iteratively adds the expert that minimizes the Jensen-Shannon divergence between the outputs of the pruned and original models, repeating this process until the desired number of experts is retained. However, for fine-grained large-scale MoE models with hundreds of experts, these methods are impractical in practice, as they require tens of thousands or even trillions of enumerations, which is obviously infeasible.

In this paper, we propose a MoE pruning method (Shapley-MoE) inspired by cooperative game theory [15, 39], which does not require enumerating all possible expert combinations and can obtain a high-performance pruned MoE model in just tens of minutes. We observe that in MoE models, experts are dynamically involved in collective decision-making through a gating mechanism that enables conditional activation. These experts function not only as independent computational units but also as collaborative combinations interacting with each other. To handle the complex relationships among experts, we leverage the Shapley value [63, 1], a key solution concept in cooperative game theory for allocating contributions to participants. By considering all possible combinations, the Shapley value quantifies each expert's contribution, thereby effectively identifying those experts that are highly relevant to task performance.

Since the exact computation of Shapley values is an NP-hard problem, we employ a Monte Carlo sampling [19, 20] approach to efficiently approximate Shapley values by sampling from the set of possible expert permutations. This successfully reduces the computational complexity from exponential to quadratic, making it feasible to compute Shapley values for experts in large-scale MoE models. However, using vanilla Monte Carlo sampling to estimate the Shapley values of MoE experts still suffers from insufficient estimation accuracy and low sampling efficiency. To enhance the accuracy and efficiency of Monte Carlo sampling, we propose the early truncation and router-

guided importance sampling (RGIS) methods. In the early truncation, we observe that sampling steps involving subsets with too few experts can lead to drastic model performance degradation, resulting in unstable sampling outcomes. We propose to terminate such sampling steps early to reduce unnecessary computation and ensure accurate Shapley value estimation. Additionally, we introduce the RGIS method. This method leverages the activation probabilities of experts from the gating mechanism, prioritizing key expert subsets with higher activation frequencies during Monte Carlo sampling, and accelerates sampling through importance weighting. With the same number of sampling steps, this method can estimate the Shapley values more accurately. We conducted theoretical analysis and ablation experiments on the two proposed methods, both of which demonstrate that these methods significantly improve the estimation accuracy and increase the sampling efficiency.

To evaluate the effectiveness of our proposed Shapley-MoE method, we conducted systematic experiments based on several mainstream open-source MoE model architectures, including the Qwen [3], DeepSeek [13], and Mixtral [35] series. The evaluation metrics cover perplexity, average zero-shot accuracy on seven tasks as well as performance on domain-specific tasks such as knowledge reasoning, arithmetic, and code generation. Experimental results show that Shapley-MoE has obvious advantages over existing MoE expert pruning methods. Specifically, when applied to the Qwen2-57B-A14B [74] model with a 25% reduction in the number of experts, its perplexity increases by only 0.92 and the average zero-shot accuracy is maintained at up to 96.4%. Furthermore, the pruned model can achieve a 1.25-2.46$\times$ reduction in GPU memory usage and a 1.26-2.92$\times$ increase in inference speed with the pruning rate ranged from 25% to 75%, while the pruning process only takes 36 minutes. Additionally, we further explored the application of the Shapley-MoE method on multimodal MoE models and performed further performance optimization on the pruned models by integrating LoRA [29] fine-tuning and quantization techniques.

## 2 Related Work

**Pruning the Expert of MoE Models.** Researchers have proposed numerous MoE pruning algorithms to remove unimportant experts. There are some works that achieve the effect of pruning MoE models through expert merging [42, 47, 78]. Other works achieve the effect of pruning MoE models by directly discarding experts. For instance, SEER-MoE [56] proposes to prune the MoE model based on the activation frequency of experts. He et al. [24] proposed to calculate the averaged routing score of each expert and retain the expert with the highest score. NAEE [49] determines the subset of retained experts by enumerating all possible combinations of experts. CD-MoE [6] introduces a greedy search algorithm that selects experts to retain based on the smallest Jensen-Shannon divergence [53]. Existing MoE pruning methods rely on heuristics or impractical enumeration. We propose Shapley value-based pruning, offering theoretical performance guarantees and efficient model pruning.

**Shapley Value.** Shapley value [23, 69] is a game theory concept that quantifies each participant's average marginal contribution across all possible coalition combinations in cooperative scenarios. Shapley value are often used to assess the importance of model components [20, 2, 71]. Neuron Shapley [20] proposed using Shapley value to quantify the contribution of individual neurons to the predictions and performance of deep networks. Ancona et al. [2] proposed Shapley value as a metric for structured pruning in convolutional neural networks. Moreover, since accurately calculating Shapley value is an NP-hard problem, many methods have been proposed to approximate Shapley value, such as Monte Carlo sampling [19, 20], kernel-based approximation algorithms [50], and stratified sampling techniques [77]. In this paper, we innovatively extend Shapley value to evaluate the importance of experts and we propose a Monte Carlo sampling approach for approximating Shapley value, which incorporates early truncation and router-guided importance sampling.

## 3 Methodology

**Notation.** In this study, we adopt the following notation conventions: bold typeface indicates vectors (*e.g.*, $x$, $y$) and matrices (*e.g.*, $X$, $Y$), while calligraphic font denotes loss function and set (*e.g.*, $\mathcal{L}, \mathcal{E}$).

## 3.1 Mixture of Experts

Given a MoE model [40, 16] comprising $L$ expert layers, let the $l$-th expert layer ($l \in \{1, 2, \ldots, L\}$) contain $n$ routing experts $\{E_1^l, E_2^l, \ldots, E_n^l\}$.[2] Each expert constitutes an independent feedforward neural network (FFN). For an input vector $\boldsymbol{x} \in \mathbb{R}^d$ with hidden dimension $d$, the output computation at the $l$-th expert layer follows:

$$\sum_{i=1}^{n} G^l(\boldsymbol{x})_i E_i^l(\boldsymbol{x}), \tag{1}$$

where $G^l(\boldsymbol{x})_i \in [0, 1]$ denotes the output of gating network for expert $i$, and $E_i^l(\boldsymbol{x})$ represents the $i$-th expert's output. The gating network is formally defined as:

$$G^l(\boldsymbol{x}) = \text{Softmax}\left(\text{TopK}\left(\boldsymbol{x}\boldsymbol{W}_g^l\right)\right), \tag{2}$$

where $\boldsymbol{W}_g^l \in \mathbb{R}^{d \times n}$ denotes the routing matrix, and $\text{TopK}(\cdot)$ operator generates a sparse routing pattern by preserving only the top-$k$ values while setting others to $-\infty$, thereby activating $k$ out of $n$ experts per input token.

## 3.2 Shapley Value of MoE Experts

**Motivation.** In MoE architectures, experts participate in collective decision-making through conditional activation governed by gating mechanism. These experts not only function as independent computational units but also form interdependent subsystems, and there is a collaborative relationship among them. To rigorously quantify individual expert contributions within this complex cooperative system, we employ Shapley Value [63, 60, 1] analysis from cooperative game theory [15, 39]. Shapley Value is a widely used mathematical tool for allocation problems, capable of fairly quantifying the marginal contribution of each participant in a collaborative process.

**Shapley Value Formulation.** The process of experts dynamically participate in decision-making through gating mechanism can be formally conceptualized as a cooperative game $\Gamma = (\mathcal{E}, V)$ where $\mathcal{E}$ denotes the expert set and $V : 2^{\mathcal{E}} \to \mathbb{R}$ represents the model performance. Perplexity is commonly used to measure the performance of LLMs and a lower perplexity indicates better model performance. Therefore, we use the reciprocal of perplexity as the metric for evaluating the performance of the MoE model, that is, $V = 1/\text{PPL}$. Each expert $E_i^l$ acts as a player whose Shapley Value [63] quantifies its average marginal contribution to model performance. Consider a MoE model with $L$ expert layers containing $n$ routing experts per layer. Let the complete expert set be $\mathcal{E} = \{E_i^l | 1 \le l \le L, 1 \le i \le n\}$ with cardinality $N = L \times n$. The Shapley Value $\phi_{E_i^l}$ for expert $E_i^l$ is defined as:

$$\phi_{E_i^l}(V) = \frac{1}{N} \sum_{\mathcal{S} \subseteq \mathcal{E} \setminus \{E_i^l\}} \frac{V(\mathcal{S} \cup \{E_i^l\}) - V(\mathcal{S})}{\binom{N-1}{|\mathcal{S}|}} \tag{3}$$

where $V(\mathcal{S})$ measures model performance when retaining subset $\mathcal{S}$. The Shapley Value $\phi_{E_i^l}$ computes the weighted average of $E_i^l$'s marginal contributions $V(\mathcal{S} \cup \{E_i^l\}) - V(\mathcal{S})$ over all $\mathcal{S} \subseteq \mathcal{E} \setminus \{E_i^l\}$. This precisely quantifies how each expert affects model performance through all possible collaborative scenarios. In addition, we present the properties of the Shapley value in Sec. C, which ensure that the contributions of MoE experts are fairly quantified across all coalition interactions.

## 3.3 Approximation of Shapley Value

**Computational Challenges.** While the Shapley value theoretically provides a metric for quantifying expert contributions, exact computation through Eq. 3 requires evaluating all $2^N$ expert subsets, which is a prohibitive proposition for modern MoE architectures where total expert count $N$ often exceeds $10^3$. This exponential complexity motivates the development of efficient approximation methods.

---

[2]Some advanced MoE architectures (*e.g.*, DeepSeekMoE-16B [13] and Qwen2-57B-A14B [73]) incorporate shared experts alongside routing experts. Our pruning strategy specifically removes less important routing experts while preserving all shared experts. For notational simplicity, we omit explicit representation of shared experts in this formulation.

**Monte Carlo Estimation.** We employ the Monte Carlo sampling approach [7] to estimate Shapley value through empirical expectation over expert permutations. For expert $E_i^l \in \mathcal{E}$, the estimator is constructed as:

$$\hat{\phi}_{E_i^l}(V) = \frac{1}{M} \sum_{m=1}^{M} \left[ V(\mathcal{S}^{(m)} \cup \{E_i^l\}) - V(\mathcal{S}^{(m)}) \right] \tag{4}$$

where $M$ denotes the number of Monte Carlo samples. Each subset $\mathcal{S}^{(m)} \subseteq \mathcal{E} \setminus \{E_i^l\}$ is generated by sampling a random permutation $\pi^{(m)}$ of all experts and taking $\mathcal{S}^{(m)} = \text{Pref}(\pi^{(m)})$, which is the set of experts preceding $E_i^l$ in permutation $\pi^{(m)}$. This stratified sampling strategy ensures uniform coverage of coalition spaces while maintaining estimator unbiasedness [62, 28]. By averaging over $M$ permutations, this method reduces the computational complexity from $\mathcal{O}(2^N)$ to $\mathcal{O}(MN)$, making it feasible for large-scale MoE models.

**Early Truncation.** During Monte Carlo sampling, we observe that subsets $\mathcal{S}^{(m)}$ with insufficient expert participation (*i.e.*, $|\mathcal{S}^{(m)}| \ll N$) frequently lead to catastrophic performance collapse ($V(\mathcal{S}^{(m)}) \to 0$), introducing high variance in Shapley value estimation. Using this fact, for a sampled permutation $\pi^{(m)}$, we can avoid computing the marginal contributions of the earlier elements. Therefore, we implement an early truncation [19] mechanism during Monte Carlo estimation. Formally, the evaluation terminates prematurely when:

$$V(\mathcal{S}^{(m)}) < \tau \cdot V(\mathcal{E}), \tag{5}$$

where $\tau \in (0, 1)$ is the threshold. This adaptive truncation achieves dual benefits: 1) It eliminates uninformative sampling steps where model functionality has collapsed, thereby reducing estimator variance. 2) It preserves computational resources by early-exiting. Our ablation studies in Sec. 4.6 show that this technique reduces the pruning cost of Shapley-MoE by nearly half and improves the accuracy of Shapley-MoE.. In addition, the error bound of early truncation is given by the following theorem:

**Theorem 1** (Error Bound of Early Truncation Shapley Estimation). *Let $\phi_{E_i^l}$ denote the true Shapley value for expert $E_i^l$ and $\hat{\phi}_{E_i^l}^{trunc}$ denote its Monte Carlo estimator with early truncation threshold $\tau$. Assume that for any subset $\mathcal{S} \subseteq \mathcal{E} \setminus \{E_i^l\}$ satisfying $V(\mathcal{S}) < \tau V(\mathcal{E})$, the marginal contribution is bounded as $|V(\mathcal{S} \cup \{E_i^l\}) - V(\mathcal{S})| \leq \epsilon$. Then, for any $\delta > 0$, with probability at least $1 - \delta$, the estimation error satisfies:*

$$|\phi_{E_i^l} - \hat{\phi}_{E_i^l}^{trunc}| \leq \epsilon \cdot \mathbb{P}(V(\mathcal{S}) < \tau V(\mathcal{E})) + \sqrt{\frac{\log(2/\delta)}{2M}} \tag{6}$$

*where $M$ is the number of Monte Carlo samples, and the probability $\mathbb{P}(V(\mathcal{S}) < \tau V(\mathcal{E}))$ is taken over the uniform distribution of expert permutations.*

The proof of the above theorem is provided in Sec. D. The theorem guarantees that early truncation preserves estimation accuracy through two mechanisms: 1) The first term $\epsilon \cdot \mathbb{P}(V(\mathcal{S}) < \tau V(\mathcal{E}))$ bounds errors from truncated subsets by their contribution probability and bounded marginal impact $\epsilon$. 2) The second term $\sqrt{\log(2/\delta)/(2M)}$ controls Monte Carlo sampling error, which diminishes with sample size $M$. The above theorem shows that the error of the early truncation is bounded, which justifies the truncation: it discards minimally influential computations while preserving estimator consistency, enabling efficient approximation.

**Router-Guided Importance Sampling.** To further accelerate the calculation of the Shapley values of the MoE experts, we further propose a Router-Guided Importance Sampling (RGIS) method based on the characteristics of the router network. This method leverages expert activation probabilities from the gating mechanism to prioritize critical coalitions during Monte Carlo estimation, achieving sampling acceleration through importance weighting. First, we profile each expert's activation frequency over a calibration dataset $\mathcal{D}$:

$$p_i^l = \frac{1}{|\mathcal{D}|} \sum_{\boldsymbol{x} \in \mathcal{D}} G^l(\boldsymbol{x})_i \tag{7}$$

where $p_i^l$ represents the empirical activation probability for expert $E_i^l$. These probabilities establish a prior distribution reflecting expert importance. Next, we generate expert permutations through

a Plackett-Luce model [58] parameterized by $\{p_i^l\}$. For each Monte Carlo sample $m$, we sample permutation $\pi^{(m)}$ by sequentially selecting experts without replacement with probability:

$$\mathbb{P}\left(\pi^{(m)}[t] = E_i^l \mid \mathcal{A}_t^{(m)}\right) = \frac{p_i^l}{\sum_{E_j^k \in \mathcal{A}_t^{(m)}} p_j^k} \tag{8}$$

where $\mathcal{A}_t^{(m)} \subseteq \mathcal{E}$ denotes the available expert set at step $t$, and $\pi^{(m)}[t]$ is the $t$-th selected expert. This sampling prioritizes high-activation experts. Finally, we compute importance weights to correct sampling bias. Let $Q(\pi^{(m)})$ denote the permutation probability under Plackett-Luce model:

$$Q(\pi^{(m)}) = \prod_{t=1}^{N} \frac{p_{\pi^{(m)}[t]}}{\sum_{s=t}^{N} p_{\pi^{(m)}[s]}} \tag{9}$$

The importance weight becomes $w^{(m)} = \frac{1/N}{Q(\pi^{(m)})}$. The RGIS Shapley estimator is:

$$\hat{\phi}_{E_i^l}^{\text{RGIS}} = \frac{1}{M} \sum_{m=1}^{M} \left[ \left( V(\mathcal{S}_i^{(m)} \cup \{E_i^l\}) - V(\mathcal{S}_i^{(m)}) \right) \cdot w^{(m)} \cdot \mathbb{I}_i^{(m)} \right] \tag{10}$$

where $\mathcal{S}_i^{(m)}$ contains experts preceding $E_i^l$ in $\pi^{(m)}$, and $\mathbb{I}_i^{(m)} \in \{0, 1\}$ indicates whether $E_i^l$ was evaluated before truncation in sample $m$.

We propose the following theorem to demonstrate the theoretical properties of the RGIS estimator. First, it guarantees that our router-guided sampling strategy does not introduce systematic estimation bias through careful importance weighting. Second, it demonstrates that leveraging router activation patterns as importance priors fundamentally reduces estimation variance compared to naive Monte Carlo sampling.

**Theorem 2** (Unbiasedness and Variance Reduction of RGIS). *The Router-Guided Importance Sampling (RGIS) estimator $\hat{\phi}_{E_i^l}^{RGIS}$ defined in Eq. 10 satisfies the following properties:*

- *Unbiasedness: For any expert $E_i^l \in \mathcal{E}$, $\mathbb{E}\left[\hat{\phi}_{E_i^l}^{RGIS}\right] \approx \phi_{E_i^l}$, where the expectation is taken over the Plackett-Luce sampling distribution $Q$.*

- *Variance Reduction: If the activation probabilities $\{p_i^l\}$ are positively correlated with the true Shapley values $\{\phi_{E_i^l}\}$, then for the same number of samples $M$, $Var\left(\hat{\phi}_{E_i^l}^{RGIS}\right) \leq Var\left(\hat{\phi}_{E_i^l}^{MC}\right)$, where $\hat{\phi}_{E_i^l}^{MC}$ denotes the standard Monte Carlo estimator in Eq. 4.*

We have provided the proof of the above theorem in Sec. E. Theorem 2 establishes that RGIS provides an unbiased Shapley value estimation while achieving variance reduction when expert activation frequencies reflect their true contributions. This occurs because the router's gating mechanism naturally prioritizes important experts, and frequently activated experts tend to have larger marginal impacts on model performance. By aligning the sampling distribution with these empirical importance measures, RGIS concentrates computational resources on evaluating the most impactful coalitions, thereby improving estimation efficiency. As shown in the ablation experiments in Sec. 4.6, RGIS improves the accuracy of Shapley-MoE. This means that it accelerates Monte Carlo sampling when given the same sampling number.

### 3.4 Shapley-MoE algorithm for MoE Pruning

We have presented our proposed Shapley-MoE for MoE expert pruning in Algorithm 1. Specifically, our Shapley-MoE algorithm prunes MoE models by estimating expert Shapley value through router-guided Monte Carlo sampling. It first profiles expert activation frequencies to prioritize critical experts during permutation sampling. For each sampled permutation, it incrementally adds experts, computes marginal performance gains, and applies early truncation when performance drops below a threshold. Finally, experts are ranked by their Shapley Values, and the lowest-ranked routing experts are pruned. Extensive experiments in Sec. 4 demonstrate that Shapley-MoE significantly outperforms current MoE pruning methods. Our method effectively reduces the memory footprint and computational cost of MoE models while maximizing the preservation of their performance.

---

**Algorithm 1** Shapley-MoE algorithm for MoE models Pruning

---

**Require:** MoE model with expert set $\mathcal{E}$, calibration dataset $\mathcal{D}$, Monte Carlo samples $M$, truncation threshold $\tau$, pruning ratio $\gamma$
**Ensure:** Pruned MoE model
1: Compute full model performance $V_{\text{full}} \leftarrow V(\mathcal{E})$ on $\mathcal{D}$
2: Profile expert activation probabilities $\{p_i^l\}$ over the $\mathcal{D}$ (Eq. 7)
3: **for** each expert $E_i^l \in \mathcal{E}$ **do**
4:    Initialize $\hat{\phi}_{E_i^l} \leftarrow 0$
5: **end for**
6: **for** $m = 1$ to $M$ **do**                                                   ▷ Monte Carlo loop
7:    Sample permutation $\pi^{(m)}$ via Plackett-Luce model with probabilities $\{p_i^l\}$ (Eq. 8)
                                                                        ▷ Router-guided importance sampling
8:    Compute permutation probability $Q(\pi^{(m)})$ and weight $w^{(m)} \leftarrow \frac{1/N}{Q(\pi^{(m)})}$ (Eq. 9)
9:    Initialize $V_0 \leftarrow V_{\text{full}}$
10:    **for** $k = 1$ to $N$ **do**                                            ▷ Permutation traversal
11:      **if** $V_{k-1} \geq \tau V_{\text{full}}$ **then**
12:        Evaluate $V_k \leftarrow V(\{\pi^{(m)}[k+1], \pi^{(m)}[k+2], ..., \pi^{(m)}[N]\})$ on $\mathcal{D}$
13:      **else**
14:        $V_k \leftarrow V_{k-1}$                                            ▷ Early truncation
15:      **end if**
16:      $\hat{\phi}_{\pi^{(m)}[k]} \leftarrow \hat{\phi}_{\pi^{(m)}[k]} + (V_{k-1} - V_k) \cdot w^{(m)}$ (Eq. 10)
17:    **end for**
18: **end for**
19: **for** each expert $E_i^l \in \mathcal{E}$ **do**
20:    $\hat{\phi}_{E_i^l} \leftarrow \hat{\phi}_{E_i^l}/M$ (Eq. 10)                        ▷ Final approximate Shapley value
21: **end for**
22: Sort routing experts by $\hat{\phi}_{E_i^l}$, remove the $\gamma$ ratio experts with the lowest ranking
23: **return** Pruned MoE model

---

## 3.5   Performance Preservation Theorem

In this section, we propose the following theorem, which establishes a performance preservation bound for our Shapley-MoE method.

**Theorem 3** (Performance Preservation Bound). *Let $\mathcal{E}$ denote the complete set of experts in the MoE model, and $\mathcal{E}_{pruned} = \mathcal{E} \setminus \{E_i^l \mid \hat{\phi}_{E_i^l} < \epsilon\}(l \in \{1, 2, \ldots, L\}, i \in \{1, 2, \ldots, n\})$ be the pruned expert set with Shapley value threshold $\epsilon > 0$. Under the following conditions:*

- *The Shapley value estimator satisfies $|\hat{\phi}_{E_i^l} - \phi_{E_i^l}| \leq \eta$ for all $E_i^l \in \mathcal{E}$ with probability $\geq 1 - \delta$.*

- *The cooperative game is additive, i.e., $V(\mathcal{S}) = \sum_{E_i^l \in \mathcal{S}} \phi_{E_i^l}$ for any $\mathcal{S} \subseteq \mathcal{E}$.*

*Then with probability at least $1 - \delta$, the performance degradation after pruning satisfies: $V(\mathcal{E}) - V(\mathcal{E}_{pruned}) \leq N(\epsilon + \eta)$, where $N = |\mathcal{E}|$ is the total number of experts.*

We have provided the proof of the above theorem in Sec. F. The above theorem provides a crucial theoretical guarantee for our method. It formally establishes that the performance degradation of the model after pruning is controllably bounded. This bound is directly linked to two practical factors: the pruning threshold ($\epsilon$) we choose and the estimation accuracy of our Shapley value calculation ($\eta$). The theorem demonstrates that by removing only experts with verifiably low contributions and ensuring a precise estimation, the overall performance loss can be provably limited, thus validating the reliability and effectiveness of our pruning methodology.

# 4 Experiments

## 4.1 Experimental Setup

**Models.** We conducted experiments on popular open-source MoE models, including the Qwen series (Qwen1.5-MoE-A2.7B [3], Qwen2-57B-A14B [73], and Qwen3-30B-A3B [68]), the DeepSeek series (DeepSeekMoE-16B [13] and DeepSeek-V2-Lite [45]), and the Mixtral series (Mixtral-8x7B [35]). The basic architecture information of these models are summarized in Sec. G.

**Baselines.** We compare strong baselines for MoE expert pruning directly related to this study, including Random [49], Gating Score [14], Frequency [56], NAEE [49], Expert Trim [24] and CD-MoE [6]. Additionally, since NAEE [49] and CD-MoE [6] determine the pruned expert set through enumeration, they are not suitable for MoE models with dozens to hundreds of experts (see analysis in Sec. H). Therefore, we only compare these two methods on Mixtral-8x7B [35].

**Benchmarks.** We follow the evaluation settings of previous MoE pruning methods to assess the zero-shot learning and language modeling capabilities of pruned MoE models. Specifically, we evaluate the zero-shot performance on seven downstream tasks: HellaSwag [76], WinoGrande [61], PIQA [4], OpenbookQA [55], ARC Easy and Challenge [12], and BoolQ [11]. The above accuracies were obtained using the EleutherAI language model evaluation framework [18]. Additionally, we evaluate the perplexity of the pruned models on the WikiText-2 [54] validation set. Furthermore, we assess the pruned MoE models' knowledge reasoning, arithmetic, and code generation capabilities, reporting 5-shot accuracy on the MMLU dataset [25], 8-shot accuracy on the GSM8K dataset [64], and 0-shot accuracy on the HumanEval dataset [8].

**Implementation Details.** We follow the settings of NAEE [49] and CD-MoE [6] method, randomly sampling 128 examples from the C4 training dataset [59] as calibration data. Additionally, we set the number of Monte Carlo sampling $M$ to 20 and the early truncation threshold $\tau$ to 0.5.

**More Models, Baselines and Benchmarks.** In Sec. I, we present additional experimental results, including pruning multimodal MoE models using our Shapley-MoE method, integrating pruned MoE models with quantization technique, further enhancing the performance of pruned MoE models through LoRA fine-tuning.

## 4.2 Zero-shot Tasks

Table 1 presents the average accuracy of pruned MoE models across seven zero-shot tasks after pruning 25% and 50% of the experts. The results clearly demonstrate that our proposed Shapley-MoE method significantly outperforms existing MoE pruning approaches. For instance, in the case of the Qwen1.5-MoE-A2.7B model with a pruning rate of 50%, the accuracy achieved by the Shapley-MoE method is 2.17% higher than that of the best Frequency method. This substantial improvement underscores the effectiveness and superiority of our approach.

Table 1: The average zero-shot accuracy across 7 tasks of pruned MoE models.

| Pruning ratio | Method | Qwen | | | DeepSeek | |
|---|---|---|---|---|---|---|
| | | 1.5-MoE-A2.7B | 2-57B-A14B | 3-30B-A3B | MoE-16B | V2-Lite |
| 0% | None | 61.70 | 65.39 | 66.36 | 61.92 | 64.16 |
| 25% | Random | 54.79 | 61.76 | 57.69 | 53.37 | 54.57 |
| | Gating Score | 55.42 | 62.16 | 58.27 | 54.23 | 55.35 |
| | Frequency | 55.63 | 62.29 | 58.68 | 54.58 | 55.90 |
| | Expert Trim | 55.16 | 62.06 | 58.27 | 53.75 | 55.44 |
| | **Shapley-MoE** | **57.10** | **63.05** | **60.02** | **56.31** | **57.29** |
| 50% | Random | 45.25 | 58.44 | 44.96 | 41.50 | 44.20 |
| | Gating Score | 46.30 | 58.98 | 46.22 | 42.51 | 45.60 |
| | Frequency | 46.84 | 59.33 | 46.54 | 42.84 | 45.96 |
| | Expert Trim | 46.35 | 59.15 | 46.27 | 42.67 | 45.85 |
| | **Shapley-MoE** | **49.01** | **60.65** | **48.28** | **44.89** | **47.08** |

## 4.3 Language Modeling

**Quantitative Evaluation.**   Table 2 shows the WikiText-2 perplexity of MoE models after pruning 25% and 50% of experts, respectively. The results indicate that all the pruned models obtained by our proposed Shapley-MoE method have significantly lower perplexity than existing MoE pruning methods. For example, for the DeepSeek-MoE-16B model with a pruning rate of 50%, the perplexity of the Shapley-MoE method is 3.17 lower than that of the best Frequency method. The above results indicate that our Shapley-MoE method not only preserves the zero-shot performance of the model well but also effectively retains the model's language modeling capability.

Table 2: The WikiText-2 perplexity of pruned MoE models.

| Pruning ratio | Method | Qwen | | | DeepSeek | |
| | | 1.5-MoE-A2.7B | 2-57B-A14B | 3-30B-A3B | MoE-16B | V2-Lite |
|---|---|---|---|---|---|---|
| 0% | None | 7.01 | 5.86 | 8.45 | 6.55 | 6.35 |
| 25% | Random | 11.01 | 7.85 | 13.72 | 10.61 | 10.52 |
| | Gating Score | 10.39 | 7.15 | 12.39 | 9.57 | 9.65 |
| | Frequency | 10.08 | 7.11 | 11.97 | 9.46 | 9.56 |
| | Expert Trim | 10.28 | 7.12 | 12.29 | 9.67 | 9.76 |
| | **Shapley-MoE** | **9.69** | **6.78** | **11.02** | **9.12** | **9.01** |
| 50% | Random | 21.87 | 9.93 | 35.74 | 37.18 | 24.92 |
| | Gating Score | 20.10 | 8.58 | 32.03 | 35.12 | 23.10 |
| | Frequency | 19.78 | 8.35 | 31.23 | 34.27 | 22.25 |
| | Expert Trim | 19.98 | 8.65 | 32.26 | 35.20 | 23.01 |
| | **Shapley-MoE** | **17.57** | **7.98** | **28.10** | **31.10** | **19.96** |

**Varying Sparsity Rates.**   In Table 3, we present the WikiText-2 perplexity performance of the pruned Mixtral-8x7B model under a wider range of pruning rates, from 12.5% to 75%. The experimental results show that across different pruning rate settings, the perplexity of the pruned models obtained by our Shapley-MoE method consistently remains lower than that of pruned models obtained by other methods, which fully demonstrates the robustness of our Shapley-MoE method.

Table 3: WikiText-2 perplexity at different pruning rates.

| Method | 12.5% | 25% | 37.5% | 50% | 62.5% | 75% |
|---|---|---|---|---|---|---|
| Random | 5.32 | 6.45 | 8.15 | 13.42 | 15.21 | 27.25 |
| Gating Score | 5.13 | 6.35 | 7.80 | 13.03 | 14.97 | 26.38 |
| Frequency | 5.10 | 6.20 | 7.75 | 12.90 | 14.83 | 26.31 |
| Expert Trim | 5.23 | 6.18 | 7.70 | 12.98 | 14.67 | 26.16 |
| NAEE | 5.02 | 6.02 | 7.71 | 12.87 | 14.46 | 26.03 |
| CD-MoE | 5.01 | 5.98 | 7.69 | 12.76 | 14.35 | 25.70 |
| **Shapley-MoE** | **4.82** | **5.60** | **7.11** | **12.01** | **13.26** | **23.86** |

Table 4: Ablation study of the effectiveness of early truncation and RGIS.

| Early Truncation | RGIS | PPL | Pruning Cost (mins) |
|---|---|---|---|
| - | - | 21.06 | 93 |
| - | ✓ | 19.24 | 92 |
| ✓ | - | 18.65 | 35 |
| ✓ | ✓ | **17.57** | **36** |

## 4.4 Knowledge Reasoning, Arithmetic and Code Generation Task

We further evaluate the broader capability retention of the pruned MoE model. This includes assessing the pruned models' performance on multi-task and cross-disciplinary language understanding and reasoning, arithmetic, and code generation. We prune 25% of the experts in the Qwen1.5-MoE-A2.7B model, and the results are presented in Table 5. Our method significantly outperforms other MoE pruning methods, further demonstrating its effectiveness in preserving the various capabilities of pruned MoE models.

Table 5: Accuracy of the pruned model on the GSM8K, MMLU and HumanEval datasets.

| Method | MMLU | GSM8K | HumanEval |
|---|---|---|---|
| None | 62.50 | 61.50 | 34.20 |
| Random | 46.20 | 24.18 | 17.34 |
| Gating Score | 47.20 | 27.14 | 18.21 |
| Frequency | 50.30 | 28.89 | 18.43 |
| Expert Trim | 49.60 | 27.09 | 17.89 |
| **Shapley-MoE** | **53.80** | **31.02** | **19.24** |

## 4.5 More Results

We provide additional experimental results in appendix. In Sec. I.1, we show the performance of pruned multimodal MoE models. In Sec. I.2, we demonstrate the performance improvements of pruned MoE models with LoRA fine-tuning. Additionally, in Sec. I.3, we combine the pruned MoE models with quantization techniques to further compress pruned model.

## 4.6 Ablation Study

**Effectivenes of Early Truncation and RGIS.**  We present an ablation study in Table 4 to evaluate the effectiveness of early truncation and RGIS, assessing the contributions of these two components to Monte Carlo sampling. Specifically, we report the WikiText-2 perplexity of the Qwen1.5-MoE-A2.7B model pruned by $50\%$ experts using Shapley-MoE. We observe that removing either early truncation or RGIS leads to degraded performance of the final pruned model. Additionally, removing early truncation increases the pruning cost. This is because early truncation eliminates sampling steps when the model function collapses, reducing the variance of the estimator and saving computational resources by exiting early, thereby accelerating the Monte Carlo sampling process. RGIS leverages the expert activation probabilities in the gating mechanism to prioritize key coalitions during Monte Carlo estimation, and achieves accelerated sampling. Under the same number of sampling steps, this allows for more accurate Shapley value estimation.

**Influence of Sampling Number $M$ and Truncation Threshold $\tau$.**  We also demonstrate the impact of the number of Monte Carlo samples $M$ and the early truncation threshold $\tau$ on the final accuracy of the pruned model. Fig. 2 shows WikiText-2 perplexity and pruning cost (minutes) of Shapley-MoE under different settings of $M$ and $\tau$. Although reducing $M$ can effectively decrease pruning cost, it may lead to performance degradation due to insufficient sampling, which affects the accuracy of Shapley value estimation. However, when the number of samples $M$ exceeds 20, the final performance of the pruned model becomes insensitive to changes in $M$. Considering both search efficiency and model performance, we ultimately set $M = 20$ as the number of samples. In addition, a moderate truncation threshold $\tau$ achieves a better trade-off between accuracy and computational complexity, as it both suppresses the variance in sampling results and further reduces pruning cost.

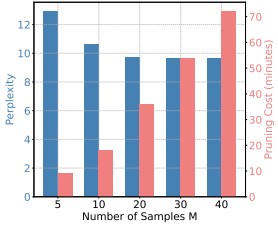
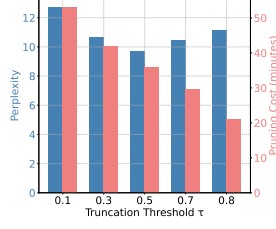
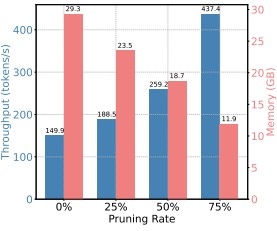

|  |  |  |
|---|---|---|
| (a) Sampling Number $M$ | (b) Truncation Threshold $\tau$ | Figure 3: Memory footprint and inference speed of pruned Qwen1.5-MoE-A2.7B. |

Figure 2: Influence of sampling number $M$ and truncation threshold $\tau$.

**Inference Speedup.**  We measure the memory reduction and inference acceleration performance of the pruned Qwen1.5-MoE-A2.7B model, with results shown in Figure 3. The results in the table are measured using the vLLM inference engine [38] on an NVIDIA A100 80GB GPU. Compared to the original model, when the pruning rate ranged from $25\%$ to $75\%$, the inference speedup is improved by $1.26\times$ to $2.92 \times$, and the memory usage on the GPU is also reduced by $1.25\times$ to $2.46 \times$.

## 4.7 More Ablation Study

We have provided additional ablation results in the appendix. Specifically, in Sec. J.1, we show the time required by Shapley-MoE to obtain different pruned models. In Sec. J.2, we analyze the impact of the calibration dataset on the accuracy of the pruned models, including the number of calibration samples and the use of domain-specific calibration datasets. Additionally, in Sec. J.3, we demonstrate the robustness of Shapley-MoE method under different random seeds.

## 5 Conclusion

This paper introduces Shapley-MoE, an efficient pruning method for MoE models that leverages Shapley value to quantify expert contributions. By integrating Monte Carlo sampling with early truncation and router-guided importance sampling, it achieves scalable and accurate pruning. Empirical results demonstrate superior performance over existing methods, enabling resource-efficient deployment of high-quality pruned MoE models.

## Acknowledgments

This work was supported by the National Science Fund for Distinguished Young Scholars (No.62025603), the National Natural Science Foundation of China (No. U21B2037, No. U22B2051, No. U23A20383, No. 62176222, No. 62176223, No. 62176226, No. 62072386, No. 62072387, No. 62072389, No. 62002305 and No. 62272401, No.624B2119), the Natural Science Foundation of Fujian Province of China (No. 2021J06003, No.2022J06001), and the Fundamental Research Funds for the Central Universities.

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

# Appendix

## A Limitations and Future Work

In this paper, we propose a novel MoE pruning framework based on the Shapley value, which effectively prunes unimportant experts, reduces memory usage, and improves inference speed. However, under high pruning ratio settings, the accuracy of the pruned MoE model drops significantly, resulting in a performance gap compared to the original model. Although using LoRA fine-tuning helps narrow this gap, there is still room for improvement to achieve high-ratio lossless pruning for MoE models. In the future, we plan to explore more efficient MoE compression techniques to further close the performance gap between compressed model and the original model under high compression ratio settings.

## B Impact Statements

This paper proposes a Shapley value-based framework for pruning MoE models. We have not found any direct negative social impacts caused by the algorithm itself. In fact, we believe that introducing our method to the community has tremendous social value. By pruning unimportant MoE experts, we can significantly reduce the number of parameters in MoE models while retaining their functionality. Therefore, this helps to reduce the computational resource consumption of MoE models and contributes to lowering carbon emissions caused by GPU computation.

## C  Properties of Shapley Value

The Shapley value satisfies four axiomatic requirements [63]:

- *Efficiency*: Total Shapley Value equal the model's full-set performance gain: $\sum_{E_i^l \in \mathcal{E}} \phi_{E_i^l} = V(\mathcal{E}) - V(\emptyset)$.
- *Symmetry*: Experts contributing equally to all coalitions receive identical values: $\phi_{E_i^l} = \phi_{E_j^k}$ if $\forall \mathcal{S} \subseteq \mathcal{E} \setminus \{E_i^l, E_j^k\}, V(\mathcal{S} \cup \{E_i^l\}) = V(\mathcal{S} \cup \{E_j^k\})$.
- *Null Player*: Experts with zero marginal impact get a Shapley value of 0: $\phi_{E_i^l} = 0$ if $\forall \mathcal{S} \subseteq \mathcal{E}, V(\mathcal{S} \cup \{E_i^l\}) = V(\mathcal{S})$.
- *Linearity*: Values are preserved under linear combinations: $\phi_{E_i^l}(\alpha V + \beta W) = \alpha \phi_{E_i^l}(V) + \beta \phi_{E_i^l}(W)$.

The above Shapley Value's axiomatic guarantees ensure expert contributions are quantified fairly across all coalitional interactions. This enables precise identification of redundant or underperforming experts (via *Null Player* property) while preserving critical collaborative experts (via *Efficiency, Symmetry and Linearity* property), forming a theoretically grounded basis for expert pruning in MoE models.

## D  Proof of Theorem 1

*Proof.* The Shapley value $\phi_{E_i^l}$ can be decomposed into contributions from two disjoint events:

$$\phi_{E_i^l} = \underbrace{\mathbb{E}[\Delta \mid \Omega] \cdot \mathbb{P}(\Omega)}_{\text{Non-truncated regions}} + \underbrace{\mathbb{E}[\Delta \mid \neg\Omega] \cdot \mathbb{P}(\neg\Omega)}_{\text{Truncated regions}}, \tag{11}$$

where $\Delta = V(\mathcal{S} \cup \{E_i^l\}) - V(\mathcal{S})$, and $\Omega$ denotes the event $V(\mathcal{S}) \geq \tau V(\mathcal{E})$. The truncated estimator $\hat{\phi}_{E_i^l}^{\text{trunc}}$ approximates the first term through Monte Carlo sampling:

$$\hat{\phi}_{E_i^l}^{\text{trunc}} = \frac{1}{M} \sum_{m=1}^{M} \Delta^{(m)} \cdot \mathbf{1}_{\Omega^{(m)}}. \tag{12}$$

The estimation error can be bounded by:

$$|\phi_{E_i^l} - \hat{\phi}_{E_i^l}^{\text{trunc}}| \leq \underbrace{|\mathbb{E}[\Delta \mid \neg\Omega] \cdot \mathbb{P}(\neg\Omega)|}_{\text{Bias term}} + \underbrace{|\mathbb{E}[\Delta \mid \Omega] \cdot \mathbb{P}(\Omega) - \hat{\phi}_{E_i^l}^{\text{trunc}}|}_{\text{Variance term}}. \tag{13}$$

*Bias Analysis:* By the theorem's assumption, $|\mathbb{E}[\Delta \mid \neg\Omega]| \leq \epsilon$. Therefore,

$$|\mathbb{E}[\Delta \mid \neg\Omega] \cdot \mathbb{P}(\neg\Omega)| \leq \epsilon \cdot \mathbb{P}(\neg\Omega). \tag{14}$$

*Variance Analysis:* The variance term corresponds to the Monte Carlo estimation error of $\mathbb{E}[\Delta \mid \Omega] \cdot \mathbb{P}(\Omega)$. Since $\Delta$ is bounded within $[-\epsilon, \epsilon]$ in the truncated regions and potentially within a larger range $[-C, C]$ in non-truncated regions (where $C = \max |V(\mathcal{S} \cup \{E_i^l\}) - V(\mathcal{S})|$), we apply Hoeffding's inequality [27]. For $M$ independent samples, with probability $\geq 1 - \delta$:

$$|\mathbb{E}[\Delta \mid \Omega] \cdot \mathbb{P}(\Omega) - \hat{\phi}_{E_i^l}^{\text{trunc}}| \leq \sqrt{\frac{C^2 \log(2/\delta)}{2M}}. \tag{15}$$

Assuming $\Delta \in [-C, C]$ for a constant $C$ (e.g., $C = V(\mathcal{E})$), the variance term is bounded by $\sqrt{\frac{\log(2/\delta)}{2M}}$ after normalizing $C$.

Combining both terms yields the stated error bound:

$$|\phi_{E_i^l} - \hat{\phi}_{E_i^l}^{\text{trunc}}| \leq \epsilon \cdot \mathbb{P}(V(\mathcal{S}) < \tau V(\mathcal{E})) + \sqrt{\frac{\log(2/\delta)}{2M}}. \tag{16}$$

$\square$

# E    Proof of Theorem 2

*Proof.* **Proof of Unbiasedness.** Let $\Pi_N$ denote the set of all permutations of the expert set $\mathcal{E}$. The true Shapley value can be expressed as:

$$\phi_{E_i^l} = \frac{1}{N} \sum_{\pi \in \Pi_N} \left[ V(\mathcal{S}_\pi^{(i)} \cup \{E_i^l\}) - V(\mathcal{S}_\pi^{(i)}) \right] \tag{17}$$

where $\mathcal{S}_\pi^{(i)}$ is the set of experts preceding $E_i^l$ in permutation $\pi$.

The RGIS estimator computes:

$$\hat{\phi}_{E_i^l}^{\text{RGIS}} = \frac{1}{M} \sum_{m=1}^{M} \left[ \left( V(\mathcal{S}^{(m)} \cup \{E_i^l\}) - V(\mathcal{S}^{(m)}) \right) \cdot w^{(m)} \cdot \mathbb{I}_i^{(m)} \right] \tag{18}$$

where $w^{(m)} = \frac{1/N}{Q(\pi^{(m)})}$ is the importance weight for permutation $\pi^{(m)}$, and $\mathbb{I}_i^{(m)}$ indicates whether $E_i^l$ was evaluated before truncation.

Taking expectation over the sampling distribution $Q$, and assuming truncation preserves unbiasedness (truncation error bounded by Theorem 1, preserving unbiasedness through controlled approximation):

$$\mathbb{E}_Q \left[ \hat{\phi}_{E_i^l}^{\text{RGIS}} \right] = \mathbb{E}_Q \left[ (V(\mathcal{S} \cup \{E_i^l\}) - V(\mathcal{S})) \cdot w \cdot \mathbb{I}_i \right]$$

$$= \sum_{\pi \in \Pi_N} Q(\pi) \cdot \left[ \left( V(\mathcal{S}_\pi^{(i)} \cup \{E_i^l\}) - V(\mathcal{S}_\pi^{(i)}) \right) \cdot \frac{1/N}{Q(\pi)} \cdot \mathbb{I}_i(\pi) \right]$$

$$= \frac{1}{N} \sum_{\pi \in \Pi_N} \left[ V(\mathcal{S}_\pi^{(i)} \cup \{E_i^l\}) - V(\mathcal{S}_\pi^{(i)}) \right] \cdot \mathbb{I}_i(\pi). \tag{19}$$

Under the truncation condition in Eq. 5, $\mathbb{I}_i(\pi)$ equals 1 for permutations where $E_i^l$ is placed before performance collapse. Since truncation only affects coalitions with negligible contributions (as shown in Theorem 1), we have:

$$\mathbb{E}_Q \left[ \hat{\phi}_{E_i^l}^{\text{RGIS}} \right] \approx \frac{1}{N} \sum_{\pi \in \Pi_N} \left[ V(\mathcal{S}_\pi^{(i)} \cup \{E_i^l\}) - V(\mathcal{S}_\pi^{(i)}) \right] = \phi_{E_i^l}. \tag{20}$$

Thus, the RGIS estimator is unbiased.

**Proof of Variance Reduction.** The variance of the standard Monte Carlo estimator is:

$$\text{Var} \left( \hat{\phi}_{E_i^l}^{\text{MC}} \right) = \frac{1}{M} \text{Var}_{\pi \sim \text{Uniform}} \left( V(\mathcal{S}_\pi^{(i)} \cup \{E_i^l\}) - V(\mathcal{S}_\pi^{(i)}) \right). \tag{21}$$

For the RGIS estimator with importance sampling:

$$\text{Var} \left( \hat{\phi}_{E_i^l}^{\text{RGIS}} \right) = \frac{1}{M} \mathbb{E}_Q \left[ \left( (V(\mathcal{S} \cup \{E_i^l\}) - V(\mathcal{S}))^2 \cdot w^2 \right) - \frac{1}{M} \phi_{E_i^l}^2 \right]$$

$$= \frac{1}{M} \sum_{\pi \in \Pi_N} \frac{\left( V(\mathcal{S}_\pi^{(i)} \cup \{E_i^l\}) - V(\mathcal{S}_\pi^{(i)}) \right)^2}{N \cdot Q(\pi)} \cdot Q(\pi) - \frac{1}{M} \phi_{E_i^l}^2$$

$$= \frac{1}{M} \cdot \frac{1}{N} \sum_{\pi \in \Pi_N} \frac{\left( V(\mathcal{S}_\pi^{(i)} \cup \{E_i^l\}) - V(\mathcal{S}_\pi^{(i)}) \right)^2}{Q(\pi)} - \frac{1}{M} \phi_{E_i^l}^2. \tag{22}$$

When activation probabilities $\{p_i^l\}$ correlate with Shapley values, the Plackett-Luce distribution $Q(\pi)$ assigns higher probability to permutations where important experts (with larger $|\phi_{E_i^l}|$) appear earlier. This makes $Q(\pi)$ inversely proportional to $\left( V(\mathcal{S}_\pi^{(i)} \cup \{E_i^l\}) - V(\mathcal{S}_\pi^{(i)}) \right)^2$ for critical experts, thereby reducing the summation term $\sum_\pi \frac{\left( V(\mathcal{S}_\pi^{(i)} \cup \{E_i^l\}) - V(\mathcal{S}_\pi^{(i)}) \right)^2}{Q(\pi)}$. Consequently:

$$\text{Var} \left( \hat{\phi}_{E_i^l}^{\text{RGIS}} \right) \leq \text{Var} \left( \hat{\phi}_{E_i^l}^{\text{MC}} \right), \tag{23}$$

completing the proof. $\qquad\square$

# F  Proof of Theorem 3

*Proof.* Under the additivity assumption,

$$V(\mathcal{E}_{\text{pruned}}) = \sum_{E_i^l \in \mathcal{E}_{\text{pruned}}} \phi_{E_i^l}. \tag{24}$$

Therefore, the performance degradation is

$$V(\mathcal{E}) - V(\mathcal{E}_{\text{pruned}}) = \sum_{E_i^l \notin \mathcal{E}_{\text{pruned}}} \phi_{E_i^l}. \tag{25}$$

For each pruned expert $E_i^l \notin \mathcal{E}_{\text{pruned}}$ (i.e., $\hat{\phi}_{E_i^l} < \epsilon$), we have

$$\phi_{E_i^l} \le \hat{\phi}_{E_i^l} + \eta < \epsilon + \eta, \tag{26}$$

with probability at least $1 - \delta$. Let $R = |\{E_i^l \notin \mathcal{E}_{\text{pruned}}\}|$ be the number of pruned experts. Then,

$$\sum_{E_i^l \notin \mathcal{E}_{\text{pruned}}} \phi_{E_i^l} < R(\epsilon + \eta) \le N(\epsilon + \eta). \tag{27}$$

Therefore we complete the proof. □

# G  Detailed Architecture Information of Different MoE Models

We have summarized the architecture information of different MoE models in Table 6, including the total number of model parameters, the number of parameters activated per token during inference, the number of routing experts in each layer, and the number of experts in each layer activated per token during inference.

Table 6: Detailed architecture information of different MoE models

| Model | Total params | Activated params | Experts num | Activated experts num |
|---|---|---|---|---|
| Qwen1.5-MoE-A2.7B | 14.3B | 2.7B | 60 | 4 |
| Qwen2-57B-A14B | 57.4B | 14.0B | 64 | 8 |
| Qwen3-30B-A3B | 30.5B | 3.3B | 128 | 8 |
| DeepSeekMoE-16B | 16.4B | 2.8B | 64 | 6 |
| DeepSeek-V2-Lite | 15.7B | 2.4B | 64 | 6 |
| Mixtral-8x7B | 46.7B | 13.0B | 8 | 2 |

# H  Analysis of Enumeration-Based MoE Pruning Methods

The previous NAEE [49] and CD-MoE [6] methods proposed determining which experts to prune based on the loss difference between the pruned MoE model and the original MoE model. They identify the optimal subset of experts by enumerating all possible combinations to achieve the best accuracy. However, in the NAEE method, it requires $C_N^{N^{\text{prune}}}$ enumerations to determine the optimal subset of experts, where $N$ and $N^{\text{prune}}$ represent the number of experts in the original and the pruned MoE models, respectively. For the CD-MoE method, it requires $(N + N^{\text{prune}}) * N/2$ enumerations. This is practically infeasible for large-scale MoE models with a large number of experts. For example, for the Qwen3-30B-A3B model with 128 experts, when pruning the model to retain 64 experts, this would require approximately $2.4 \times 10^{37}$ and $12,288$ enumerations for the NAEE and CD-MoE methods, respectively. In contrast, our Shapley-MoE method can obtain the pruned MoE model within just 36 minutes, which is highly efficient.

# I  More Results

## I.1  Multimodal Tasks

We further apply our Shapley-MoE method to prune multimodal MoE models, demonstrating the applicability of our approach in multimodal scenarios. Specifically, we use our approach to prune the MoE-LLaVA-Qwen-1.8B-4e model [43], a multimodal MoE model with 12 MoE layers, each containing 4 experts, where only 2 experts are dynamically activated per token. The total parameter count of the MoE-LLaVA-Qwen-1.8B-4e model is 3.1B, with 2.2B parameters dynamically activated per token. We prune 25% of the model's experts and evaluate the performance of the pruned model on various visual question answering and reasoning benchmarks, including VQAv2 [21], GQA [34] and ScienceQA-IMG [48]. The results in Table 7 show that our method outperforms other MoE pruning methods, achieving improvements of 1.7%, 2.0%, and 2.4% over the best-performing method on the VQAv2, GQA and ScienceQA-IMG datasets, respectively.

Table 7: Performance of the MoE-LLaVA-Qwen-1.8B-4e model with 25% experts pruned on VQA, VQAv2 and ScienceQA-IMG datasets.

| Method | VQAv2 | GQA | ScienceQA-IMG |
|---|---|---|---|
| None | 76.20 | 61.50 | 63.10 |
| Random | 66.90 | 51.20 | 52.70 |
| Gating Score | 68.10 | 53.90 | 54.70 |
| Frequency | 68.40 | 54.70 | 55.80 |
| Expert Trim | 67.80 | 54.30 | 55.10 |
| **Shapley-MoE** | **70.10** | **56.70** | **58.20** |

## I.2  Using LoRA to Fine-tune the Pruned MoE Models

To address the notable accuracy degradation observed in pruned MoE model under high pruning rates, we further validate the effectiveness of applying LoRA [29] fine-tuning to mitigate the accuracy gap between highly pruned MoE model and their original counterpart. Specifically, we select 1000 samples from the Alpaca-GPT4 [57] dataset to fine-tune the Qwen1.5-MoE-A2.7B model, which has undergone pruning with 50% of its experts removed. During fine-tuning, we set the LoRA rank to 8. We evaluate the perplexity and average zero-shot accuracy of pruned MoE model generated via different pruning methods, both with and without LoRA fine-tuning. As shown in Table 8, LoRA fine-tuning significantly enhances the accuracy of the pruned MoE model, effectively narrowing the performance gap between the pruned model and the original model. Furthermore, since the pruned model produced by our proposed Shapley-MoE method demonstrate superior accuracy, they exhibit a greater capacity for performance recovery through LoRA fine-tuning. Notably, the performance advantage of our Shapley-MoE method persists even after fine-tuning.

Table 8: The comparison of perplexity and average zero-shot accuracy of the Qwen1.5-MoE-A2.7B model after pruning 50% experts and performing LoRA fine-tuning.

| Method | Fine-tuning | Perplexity ($\downarrow$) | Accuracy ($\uparrow$) |
|---|---|---|---|
| None | N.A. | 7.01 | 61.70 |
| Random | N.A. | 21.87 | 45.25 |
| Random | LoRA | 13.66 | 50.24 |
| Gating Score | N.A. | 20.10 | 46.30 |
| Gating Score | LoRA | 12.90 | 51.20 |
| Frequency | N.A. | 19.78 | 46.84 |
| Frequency | LoRA | 12.60 | 51.45 |
| Expert Trim | N.A. | 19.98 | 46.35 |
| Expert Trim | LoRA | 12.97 | 51.21 |
| **Shapley-MoE** | **N.A.** | **17.57** | **49.01** |
| **Shapley-MoE** | **LoRA** | **11.04** | **53.20** |

## I.3 Integrate with Quantization Technique

We further demonstrate the additional benefits of compression and acceleration brought by quantizing the pruned MoE model. Specifically, we first prune 25% experts of the Qwen1.5-MoE-A2.7B model and then apply the AWQ [44] method to quantize the pruned model to 4 bits. We also measure the memory usage and inference speed of the compressed model on an NVIDIA A100 80GB GPU. The results in Table 9 and Table 10 show that quantizing the pruned model to 4 bits further reduces memory usage and improves inference speed. Additionally, it is worth noting that the pruned model obtained by our method maintains optimal performance even after quantization, which indicates that better-performing pruned models are more compatible with other compression techniques, resulting in superior compression models.

Table 9: The WikiText-2 perplexity performance of the Qwen1.5-MoE-A2.7B model after combining the pruning and quantization technique.

| Pruning Method | bits | PPL |
|---|---|---|
| Random | 16 | 11.01 |
| W/AWQ | 4 | 12.04 |
| Gating Score | 16 | 10.39 |
| W/AWQ | 4 | 11.12 |
| Frequency | 16 | 10.08 |
| W/AWQ | 4 | 10.89 |
| Expert Trim | 16 | 10.28 |
| W/AWQ | 4 | 11.02 |
| **Shapley-MoE** | **16** | **9.69** |
| **W/AWQ** | **4** | **10.34** |

Table 10: The GPU memory usage and inference speed of quantized pruned Qwen1.5-MoE-A2.7B model.

| bits | Memory (GB) | Reduction ↓ | Throughput (tokens/s) | Speedup ↑ |
|---|---|---|---|---|
| Unpruned ×16 bit | 29.30 | 1.00× | 149.90 | 1.00× |
| Pruned × 16 bit | 23.50 | 1.25× | 188.50 | 1.26× |
| **Pruned × 4 bit** | **7.70** | **4.13×** | **593.80** | **3.96×** |

## J More Ablation Study

### J.1 Pruning Efficiency

In Table 11, we report the time required to prune different MoE models using our Shapley-MoE method. The results were obtained on NVIDIA A100 80GB GPUs. For MoE models with different parameter sizes, Shapley-MoE method only takes tens of minutes to obtain the pruned model. This demonstrates that our Shapley-MoE method is highly efficient and can quickly produce pruned MoE models. Furthermore, we believe that the efficiency advantage of our pruning method will remain for MoE models with even larger parameter sizes.

Table 11: Pruning cost of our Shapley-MoE method (in minutes).

| Model | Qwen1.5-MoE -A2.7B | Qwen2-57B -A14B | Qwen3-30B -A3B | DeepSeekMoE -16B | DeepSeek -V2-Lite | Mixtral -8x7B |
|---|---|---|---|---|---|---|
| Cost | 36 | 112 | 172 | 50 | 40 | 42 |

### J.2 Calibration Dataset

**Number of Calibration Samples.** We illustrate the impact of the number of calibration samples on the WikiText-2 perplexity of the pruned MoE model in Figure 4. Specifically, we present the perplexity of the Qwen1.5-MoE-A2.7B model with 25% of experts pruned. We observe that as the

number of calibration samples increases, the perplexity of the pruned model decreases gradually. However, when the number of calibration samples reaches 256, the perplexity of the model does not decrease further. Considering that increasing calibration samples leads to longer pruning process time and the perplexity gain brought by the increase in calibration samples, we choose 128 samples for calibration.

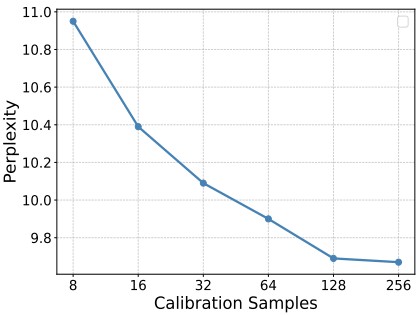

Figure 4: The WikiText-2 perplexity of the Qwen1.5-MoE-A2.7B model with 25% experts pruned under different calibration samples.

**Domain-specific Calibration Datasets.** We investigate the impact of using domain-specific datasets for calibration on the performance of pruned models in specialized domains. Specifically, We construct a domain-specific calibration dataset containing 128 examples by randomly sampling sentences from the MATH [26] and CodeQA [46] training sets. These datasets represent the arithmetic and code-generation domains and are used to calibrate the pruned Qwen1.5-MoE-A2.7B model, in which 25% of the experts have been pruned. The performance of the pruned model is subsequently evaluated on the GSM8K dataset for 8-shot accuracy and the HumanEval dataset for 0-shot accuracy, with the results summarized in Table 12. The results indicate that, compared to using the C4 dataset for pruning model calibration, employing domain-specific calibration datasets yields notable improvements in the pruned model's accuracy on the GSM8K and HumanEval tasks. This indicates that when handling tasks in specific domains, using datasets designed for these specific tasks for calibration can achieve better MoE model pruning results compared to using general pretraining datasets.

Table 12: Accuracy of the Qwen1.5-MoE-A2.7B model with 25% experts pruned on the GSM8K and HumanEval datasets when using domain-specific calibration datasets.

| Calibration | GSM8K | Calibration | HumanEval |
|---|---|---|---|
| Unpruned model | 61.50 | Unpruned model | 34.20 |
| C4 | 31.02 | C4 | 19.24 |
| **MATH** | **43.76** | **CodeQA** | **26.56** |

### J.3   Robustness of Shapley-MoE under Different Random Seeds

To demonstrate the robustness of our method, we report the WikiText2 perplexity of the Qwen1.5-MoE-A2.7B model pruned by 50% experts using Shapley-MoE across five random seeds and different calibration sets in Table 13. From the experimental results, we can observe that the variance among different random seeds is very low, and Shapley-MoE consistently outperforms other MoE pruning algorithms for different random seeds, indicating the robustness of Shapley-MoE.

## K   Detailed Results for Zero-shot Tasks

In this section, we present the detailed results of the zero-shot accuracy of the pruned MoE model on seven tasks.

Table 13: WikiText2 perplexity of the Qwen1.5-MoE-A2.7B model with 50% experts pruned across five random seeds.

| Method | Perplexity |
|---|---|
| Random | 21.87 ($\pm$) 0.13 |
| Gating Score | 20.10 ($\pm$) 0.12 |
| Frequency | 19.78 ($\pm$) 0.08 |
| Expert Trim | 19.98 ($\pm$) 0.08 |
| **Shapley-MoE** | **17.57 ($\pm$) 0.06** |

Table 14: Zero-shot task results of MoE models with 25% experts pruned.

| Model | Method | HellaSwag | Winogrande | BoolQ | OBQA | PIQA | ARC-e | ARC-c | Mean |
|---|---|---|---|---|---|---|---|---|---|
| Qwen1.5-MoE-A2.7B | None | 57.92 | 68.75 | 79.78 | 30.40 | 80.09 | 73.06 | 41.89 | 61.70 |
| | Random | 51.20 | 62.38 | 71.35 | 25.60 | 72.88 | 63.98 | 36.11 | 54.79 |
| | Gating Score | 52.10 | 63.01 | 72.41 | 25.80 | 73.01 | 64.35 | 37.24 | 55.42 |
| | Frequency | 52.21 | 63.10 | 71.90 | 25.90 | 73.19 | 65.31 | 37.78 | 55.63 |
| | Expert Trim | 51.89 | 63.12 | 71.90 | 25.00 | 73.20 | 63.90 | 37.14 | 55.16 |
| | **Shapley-MoE** | **53.90** | **65.68** | **74.89** | **27.00** | **74.09** | **65.01** | **39.12** | **57.10** |
| Qwen2-57B-A14B | None | 62.86 | 74.03 | 86.39 | 32.60 | 80.96 | 74.75 | 46.16 | 65.39 |
| | Random | 57.86 | 70.92 | 82.74 | 30.20 | 76.89 | 71.53 | 42.20 | 61.76 |
| | Gating Score | 58.16 | 71.91 | 82.89 | 30.40 | 76.98 | 71.98 | 42.81 | 62.16 |
| | Frequency | 58.02 | 72.25 | 83.10 | 30.50 | 77.05 | 72.45 | 42.65 | 62.29 |
| | Expert Trim | 57.78 | 71.57 | 82.97 | 30.70 | 77.05 | 72.18 | 42.16 | 62.06 |
| | **Shapley-MoE** | **59.15** | **72.97** | **84.56** | **31.00** | **78.12** | **72.56** | **43.02** | **63.05** |
| Qwen3-30B-A3B | None | 59.58 | 70.24 | 88.62 | 34.40 | 79.60 | 79.25 | 52.82 | 66.36 |
| | Random | 49.52 | 62.77 | 81.87 | 28.60 | 69.46 | 69.24 | 42.43 | 57.69 |
| | Gating Score | 50.23 | 62.98 | 82.18 | 29.20 | 70.03 | 70.23 | 43.01 | 58.27 |
| | Frequency | 50.68 | 63.20 | 82.56 | 29.60 | 70.27 | 70.56 | 43.90 | 58.68 |
| | Expert Trim | 50.64 | 63.37 | 82.01 | 29.10 | 69.89 | 69.79 | 43.09 | 58.27 |
| | **Shapley-MoE** | **52.35** | **65.01** | **84.12** | **30.10** | **71.56** | **71.23** | **45.80** | **60.02** |
| DeepSeekMoE-16B | None | 58.09 | 70.40 | 72.91 | 32.20 | 78.67 | 75.84 | 45.31 | 61.92 |
| | Random | 46.42 | 64.04 | 69.46 | 24.80 | 70.33 | 63.60 | 34.92 | 53.37 |
| | Gating Score | 47.09 | 65.35 | 70.56 | 25.30 | 71.26 | 64.32 | 35.75 | 54.23 |
| | Frequency | 47.67 | 66.09 | 71.05 | 25.40 | 71.87 | 64.09 | 35.90 | 54.58 |
| | Expert Trim | 46.45 | 64.95 | 69.89 | 24.80 | 70.43 | 63.89 | 35.86 | 53.75 |
| | **Shapley-MoE** | **50.13** | **67.15** | **71.45** | **28.00** | **72.09** | **66.45** | **38.90** | **56.31** |
| DeepSeek-V2-Lite | None | 58.68 | 71.35 | 79.72 | 34.20 | 80.14 | 78.37 | 46.67 | 64.16 |
| | Random | 47.48 | 66.06 | 72.60 | 25.00 | 70.89 | 64.65 | 35.32 | 54.57 |
| | Gating Score | 49.02 | 66.25 | 72.90 | 25.40 | 71.67 | 64.91 | 37.31 | 55.35 |
| | Frequency | 50.21 | 67.24 | 73.00 | 25.10 | 71.98 | 65.78 | 38.02 | 55.90 |
| | Expert Trim | 49.43 | 66.89 | 73.24 | 24.30 | 70.91 | 65.29 | 38.00 | 55.44 |
| | **Shapley-MoE** | **51.20** | **67.29** | **76.01** | **26.30** | **73.10** | **67.67** | **39.45** | **57.29** |

Table 15: Zero-shot task results of MoE models with 50% experts pruned.

| Model | Method | HellaSwag | Winogrande | BoolQ | OBQA | PIQA | ARC-e | ARC-c | Mean |
|---|---|---|---|---|---|---|---|---|---|
| Qwen1.5-MoE-A2.7B | None | 57.92 | 68.75 | 79.78 | 30.40 | 80.09 | 73.06 | 41.89 | 61.70 |
| | Random | 40.27 | 57.72 | 60.25 | 18.80 | 62.49 | 49.88 | 27.33 | 45.25 |
| | Gating Score | 42.90 | 58.93 | 61.65 | 19.40 | 62.90 | 50.28 | 28.01 | 46.30 |
| | Frequency | 43.88 | 59.46 | 61.90 | 19.80 | 63.37 | 50.80 | 28.69 | 46.84 |
| | Expert Trim | 42.98 | 58.57 | 61.89 | 19.10 | 62.96 | 50.57 | 28.35 | 46.35 |
| | **Shapley-MoE** | **45.78** | **60.90** | **65.09** | **21.50** | **64.97** | **53.96** | **30.89** | **49.01** |
| Qwen2-57B-A14B | None | 62.86 | 74.03 | 86.39 | 32.60 | 80.96 | 74.75 | 46.16 | 65.39 |
| | Random | 53.57 | 68.13 | 78.78 | 27.20 | 73.97 | 67.51 | 39.87 | 58.44 |
| | Gating Score | 54.13 | 69.14 | 79.34 | 27.60 | 74.23 | 68.14 | 40.26 | 58.98 |
| | Frequency | 54.46 | 69.45 | 79.67 | 27.90 | 74.67 | 68.56 | 40.57 | 59.33 |
| | Expert Trim | 54.36 | 69.01 | 79.02 | 27.30 | 74.87 | 68.65 | 40.87 | 59.15 |
| | **Shapley-MoE** | **56.78** | **70.92** | **81.13** | **28.90** | **75.78** | **69.18** | **41.89** | **60.65** |
| Qwen3-30B-A3B | None | 59.58 | 70.24 | 88.62 | 34.40 | 79.60 | 79.25 | 52.82 | 66.36 |
| | Random | 36.36 | 56.06 | 66.16 | 20.00 | 58.68 | 48.78 | 28.69 | 44.96 |
| | Gating Score | 38.89 | 56.89 | 68.98 | 20.70 | 58.87 | 49.56 | 29.65 | 46.22 |
| | Frequency | 39.45 | 57.90 | 67.90 | 20.80 | 58.98 | 49.79 | 30.98 | 46.54 |
| | Expert Trim | 38.87 | 57.10 | 69.12 | 20.90 | 58.78 | 49.16 | 29.98 | 46.27 |
| | **Shapley-MoE** | **41.95** | **59.12** | **71.23** | **22.30** | **60.14** | **51.24** | **31.98** | **48.28** |
| DeepSeekMoE-16B | None | 58.09 | 70.40 | 72.91 | 32.20 | 78.67 | 75.84 | 45.31 | 61.92 |
| | Random | 33.67 | 55.35 | 58.93 | 16.00 | 59.55 | 43.49 | 23.49 | 41.50 |
| | Gating Score | 34.67 | 56.36 | 60.01 | 17.10 | 60.12 | 44.23 | 25.09 | 42.51 |
| | Frequency | 33.90 | 57.76 | 60.13 | 18.00 | 59.89 | 44.21 | 26.00 | 42.84 |
| | Expert Trim | 34.56 | 57.01 | 59.79 | 18.10 | 59.81 | 43.13 | 26.29 | 42.67 |
| | **Shapley-MoE** | **36.98** | **59.01** | **61.34** | **19.30** | **62.02** | **46.67** | **28.90** | **44.89** |
| DeepSeek-V2-Lite | None | 58.68 | 71.35 | 79.72 | 34.20 | 80.14 | 78.37 | 46.67 | 64.16 |
| | Random | 34.72 | 57.41 | 63.31 | 21.80 | 60.53 | 47.11 | 24.51 | 44.20 |
| | Gating Score | 36.10 | 59.10 | 64.57 | 22.30 | 61.94 | 49.01 | 26.19 | 45.60 |
| | Frequency | 37.64 | 59.57 | 65.08 | 23.00 | 62.45 | 48.45 | 25.56 | 45.96 |
| | Expert Trim | 38.58 | 59.21 | 65.12 | 23.10 | 61.23 | 48.67 | 25.06 | 45.85 |
| | **Shapley-MoE** | **39.45** | **60.34** | **67.13** | **24.00** | **62.86** | **49.05** | **26.76** | **47.08** |

