# OpenReview forum: "Discovering Important Experts for Mixture-of-Experts Models Pruning Through a Theoretical Perspective"
_NeurIPS.cc/2025/Conference — NeurIPS 2025 poster_

### Official Review · Reviewer_vwFD · 2025-06-21

**Clarity:** 3
**Significance:** 1
**Originality:** 2
**Rating:** 4
**Confidence:** 4

**Summary:**

This paper proposes a metric (namely, Shapley value) for measuring the relative importance of the experts in Mixture-of-Experts (MoE) models, to prune unimportant experts from the model. Due to the large computational complexity of the exact evaluation of the proposed metric, the paper introduces a Monte Carlo-based sampling strategy to approximate the metric for each expert. Theoretical support has been provided for the proposed sampling strategy, along with empirical evidence.

**Questions:**

1. I am confused about the results in Theorem 1. With τ→0 (i.e., no early truncation), the first term on the right side of equation (6) diminishes, i.e., we have a tighter bound. Then, how does the early truncation help, as we are having a looser bound for the case?

2. In equation (7), over which distribution is $\mathbb{E}[G(x)]$ calculated?

**Ethical Concerns:**

["NO or VERY MINOR ethics concerns only"]

**Final Justification:**

The authors clarified a major part of the weaknesses. Therefore, I updated the rating.

**Limitations:**

Yes

**Paper Formatting Concerns:**

I did not notice any major formatting issues.

**Quality:**

2

**Strengths And Weaknesses:**

Strengths:

1.	The proposed method outperforms other baselines evaluated in this paper
2.	In terms of model and task diversity, the empirical results are adequate
3.	The paper is well-written and easy to follow

Weaknesses:

1.	The designed metric (Shapley value) does not seem completely novel to me. Although the metric considers all possible combinations of unpruned experts when evaluating the importance of a given expert, it still measures importance based on the change in model performance when that expert is pruned versus retained, using a calibration dataset—similar to the approach in [1]. The consideration of all possible combinations of unpruned experts while evaluating the metric is to capture the interdependency of the experts. However, no empirical evidence is provided to justify that there exists a sufficient level of interdependency. Moreover, it is not clear whether the interdependency is inter-layer or intra-layer.

2.	The theoretical results are only to support the sampling strategy to approximate the metric. However, no theoretical support is provided to show that the metric is itself effective.

3.	The paper compared the proposed method’s performance against some baselines. Similar to the proposed method, all those baselines are calibration data dependent, either evaluating the sensitivity of the individual experts over the calibration dataset or using the routing statistics (e.g., Gating score, Frequency) on the dataset. Comparison with data-free approach as in [2] would better evaluate the effectiveness of the proposed method.

[1] Lu, Xudong, et al. "Not All Experts are Equal: Efficient Expert Pruning and Skipping for Mixture-of-Experts Large Language Models." Proceedings of the 62nd Annual Meeting of the Association for Computational Linguistics (Volume 1: Long Papers). 2024.

[2] Chowdhury, Mohammed Nowaz Rabbani, et al. "A Provably Effective Method for Pruning Experts in Fine-tuned Sparse Mixture-of-Experts." International Conference on Machine Learning. PMLR, 2024.

---

> ### Author Rebuttal · Authors · 2025-07-31
>
> **Thanks for your careful review and comments!**
>
> > W1: The designed metric (Shapley value) does not seem completely novel to me. Although the metric considers all possible combinations of unpruned experts when evaluating the importance of a given expert, it still measures importance based on the change in model performance when that expert is pruned versus retained, using a calibration dataset—similar to the approach in [1]. The consideration of all possible combinations of unpruned experts while evaluating the metric is to capture the interdependency of the experts. However, no empirical evidence is provided to justify that there exists a sufficient level of interdependency. Moreover, it is not clear whether the interdependency is inter-layer or intra-layer.
>
> Thank you for your thoughtful feedback. We appreciate the opportunity to clarify and strengthen our contributions.
> 1. **Novelty of Shapley value metric and comparison with [1]**:
>     - Although both NAEE and our method measures importance based on the change in model performance when that expert is pruned versus retained, NAEE [1] enumerates all possible combinations of retained experts and selects the combination that minimizes the Frobenius norm of the output difference before and after pruning. **For fine-grained large-scale MoE with hundreds of experts, NAEE [1] method are impractical in practice, as they require tens of thousands or even trillions of enumerations, which is obviously infeasible.**
>     - **On the contrary, our method does not require enumerating all possible expert combinations and can obtain a high-performance pruned MoE in just tens of minutes.** By quantifying the contribution of each expert through Shapley value, thereby effectively identifying those experts that are highly relevant to task performance. In addition, we use Monte Carlo sampling to approximate Shapley value. **Crucially, we have improved the accuracy and efficiency of this approximation through novel early truncation and Router-Guided Importance Sampling (RGIS) techniques. These constitute the core algorithmic innovations we propose.** Therefore, our method has significant novelty compared to previous approaches.
> 2. **Empirical evidence for expert interdependencies:** We acknowledge that direct evidence for the level of interdependency was not explicitly quantified in our paper. We believe that the experts in MoE naturally have interdependent relationships, which include both intra-layer and inter-layer relationships. Intra-layer interdependency occur when experts within a layer compete/cooperate via top-k routing, while inter-layer ones arise from sequential processing (e.g., an expert in layer $l$ relying on outputs influenced by experts in layer $l-1$). **Therefore, our formulation treats the expert set $\mathcal{E}$ globally (across all layers), naturally capturing inter-layer and intra-layer interdependency.** For empirical evidence, please see our superior performance over baselines that do not model interdependencies (e.g., Frequency and Gating Score in Tables 1, 2 and 3). This gap indirectly evidences interdependencies, as these metrics underperform by pruning "important" experts in isolation without considering model interdependencies.
>
> > W2: The theoretical results are only to support the sampling strategy to approximate the metric. However, no theoretical support is provided to show that the metric is itself effective.
>
> The effectiveness of Shapley value itself as a metric rests on its axiomatic guarantees from cooperative game theory, which we have shown in Section C. These properties ensure it is uniquely suited for expert contribution assessment in MoE:
> 1. **Efficiency** ensures the sum of Shapley values equals the total performance gain achievable by the expert set. Pruning low Shapley value experts thus minimizes degradation, as their absence has little impact on the overall utility.
> 2. **Symmetry** ensures equitable treatment of experts with identical impacts, preventing bias in pruning decisions.
> 3. **Null Player Property** identifies experts with zero marginal impact (Shapley value is 0), enabling safe pruning without performance loss.
> 4. **Linearity** preserves the metric's validity under combinations of games, which is relevant for multi-layer MoE where performance can be viewed as additive across layers.
>
> These axioms collectively establish Shapley Value as a theoretically grounded metric for fair attribution in cooperative settings and ensure its effectiveness.
>
> > W3: The paper compared the proposed method’s performance against some baselines. Similar to the proposed method, all those baselines are calibration data dependent, either evaluating the sensitivity of the individual experts over the calibration dataset or using the routing statistics (e.g., Gating score, Frequency) on the dataset. Comparison with data-free approach as in [2] would better evaluate the effectiveness of the proposed method.
>
> 1. **Comparison with [2].**
>     - [2] proposes to prune experts based on the change in the $l_2$ norm of the router in a fine-tuned MoE compared to that in pre-trained MoE. For example, when pruning a vision MoE fine-tuned on CIFAR-10, the pruning metric is $l_2$ norm difference of the router between this fine-tuned model and the model pre-trained on ImageNet-21k. It should be noted that although this pruning process does not require a dataset, the process of obtaining the fine-tuned model using a fine-tuning dataset does require a dataset. **Therefore, strictly speaking, the method proposed in [2] is not a data-free method. In addition, the method proposed in [2] is not applicable to most scenarios of pruning Mixture-of-Experts LLMs, because there is no corresponding pre-trained model for pruning in such cases. In contrast, our method can directly prune any MoE without comparing the $l_2$ norm difference with the pre-trained model.**
>     - The method proposed in [2] cannot be used to prune MoE model mentioned in our paper, so it cannot be directly compared with our method. We migrated our method to prune the V-MoE model mentioned in [2]. We used the training set of CIFAR-10 as the calibration dataset, evaluated CIFAR-10 test set accuracy, and pruned 50% experts without fine-tuning after pruning. The results are as follows:
> |Method|Accuracy|
> |-|-|
> |Method in [2]|98.13%|
> |**Ours**|**98.32%**|
>
>         **Our method outperforms the method proposed in [2], which once again demonstrates the effectiveness of our method.** We will include the above comparison results with the data-free approach [2] in the final version of our paper.
>
> Furthermore, we would like to make the following clarifications regarding the calibration dataset.
> 1. **The use of a calibration dataset is a common practice in expert pruning for MoE.** Our method does require the use of a calibration dataset, but the use of a calibration dataset is a common practice in expert pruning for MoE. All the expert pruning methods compared in our paper, including Gating Score, Frequency, NAEE, Expert Trim, and CD-MoE, adopt calibration datasets.
>
> 2. **The use of calibration datasets is also a common practice in compressing LLMs.** Many previous SOTA methods for LLMs pruning (e.g., SparseGPT [A1], and LLM-Pruner [A2]) and quantization (e.g., AWQ [A3] and SmoothQuant [A4]) have employed calibration datasets. By using calibration dataset and running the forward propagation of the model, we can measure the weights that have a significant impact on the final output of the model, thereby reducing compression error.
>
> [A1] SparseGPT: Massive Language Models Can be Accurately Pruned in One-Shot. ICML 2023.
>
> [A2] LLM-Pruner: On the Structural Pruning of Large Language Models. NeurIPS 2023.
>
> [A3] AWQ: Activation-aware Weight Quantization for On-Device LLM Compression and Acceleration. MLSys 2024.
>
> [A4] SmoothQuant: Accurate and Efficient Post-Training Quantization for Large Language Models. ICML 2023.
>
> > Q1: With $\tau→0$ (i.e., no early truncation), the first term on the right side of equation (6) diminishes, i.e., we have a tighter bound. Then, how does the early truncation help, as we are having a looser bound for the case?
>
> 1. It is correct that the first term in the error bound on the right side of equation 6 approaches zero as $\tau→0$ (approaching no truncation), resulting in a tighter theoretical bound that reduces to the standard Monte Carlo sampling error term. However, the primary purpose of Theorem 1 is not to claim that early truncation tightens the bound compared to no truncation, but rather to provide a rigorous guarantee that the truncated estimator remains accurate (with bounded error) despite skipping certain low-information sampling steps.
> 2. The key benefit of early truncation is computational efficiency, which indirectly leads to improved estimation accuracy in practice. Without truncation, each Monte Carlo sample requires evaluating the full permutation of $N$ experts, leading to $\mathcal{O}(MN)$ complexity that is often prohibitive. Truncation allows us to early-exit permutations where performance has collapsed, significantly reducing the average number of evaluations per sample. This enables us to afford a larger number of samples $M$ within the same computational budget, which in turn reduces the second term in the bound (the sampling error). As shown in our ablation studies (Section 4.6), this technique reduces the overall pruning cost by nearly half while improving accuracy.
>
> > Q2: In equation (7), over which distribution is $\mathbb{E}(G(\boldsymbol{x}))$ calculated?
>
> Thank you for your careful reading and this was an oversight on our part. $\mathbb{E}(G(\boldsymbol{x}))$ is not calculated over any distribution and the expectation symbol $\mathbb{E}$ is redundant, and we will remove $\mathbb{E}$ in the final version of the paper.
>
> **We hope that our response has addressed your concerns, and we kindly ask for a more favorable evaluation of our work.**

---

> ### Comment · Reviewer_vwFD · 2025-08-05
>
> Thank you for your rebuttal. My replies are given below:
>
> 1. Regarding Q1: Thank you for your clarification. I was confused (didn't count this as a weakness) about the results of Theorem 1. Your answer clarifies my confusion.
>
> 2. Regarding W1, W2, and W3: I am fully aware of using the calibration dataset in previous works for compressing LLMs in general and pruning experts in MoE in particular. For the same reason, evaluating the sensitivity of the experts on the calibration dataset (certainly, in a different way using the Shapley value to improve the performance and then sampling strategy to reduce the computational overhead) still seems to lack novelty. One of the major claims of this work is to reduce the computational overhead of finding prunable experts. That is why I asked the authors to compare the results with the router-norm-based method. The authors claim that the router-norm-based method is not data-free, as it uses finetuning dataset. However, my view regarding this work is different from the authors', as here the fine-tuning dataset is used for training (which indeed requires a sufficient amount of data), not for identifying the sensitivity of the experts. The authors provided a comparison with the specified method. However, improving the results by only 0.20% with the additional complexity and computational overhead does not seem a major advantage. Also, the authors did not compare for other pruning ratios. Finally, I still believe that providing some sort of theoretical performance guarantee of the pruned MoE model itself while using the Shapley value would greatly benefit the work.
>
> I greatly appreciate the effort of the authors during the rebuttal.

---

> > ### Author Response · Authors · 2025-08-07
> > **Response to Reviewer vwFD (1/3)**
> >
> > **Thank you for your further response.**
> >
> > > For the same reason, evaluating the sensitivity of the experts on the calibration dataset (certainly, in a different way using the Shapley value to improve the performance and then sampling strategy to reduce the computational overhead) still seems to lack novelty.
> >
> > Thank you for your comments, and we would like to clarify your concerns about the novelty of our approach.
> > 1. **Comparison with previous pruning methods for MoE models.**
> >    - Although both our method and NAEE [1] involve evaluating the sensitivity of experts on the calibration dataset, and both measure importance based on the change in model performance when an expert is pruned versus retained, NAEE [1] enumerates all possible combinations of retained experts and selects the combination that minimizes the Frobenius norm of the output difference before and after pruning. For fine-grained large-scale MoE models with hundreds of experts, NAEE [1] is impractical in practice, as they require tens of thousands or even trillions of enumerations, which is obviously infeasible.
> >    - **In this paper, we propose a MoE pruning method inspired by cooperative game theory, which does not require enumerating all possible expert combinations and can obtain a high-performance pruned MoE model in just tens of minutes. Compared to NAEE [1] which uses the Frobenius norm of the output difference before and after pruning to obtain the pruned MoE model, we use the Shapley value to measure the importance of experts. Therefore, our approach and NAEE [1] adopt different pruning metrics.** Our approach draws from cooperative game theory to fairly quantify each expert's marginal contribution across all possible coalitions of experts. By modeling the MoE architecture as a cooperative game, Shapley value captures the interdependent and collaborative nature of experts, ensuring that importance scores reflect not just isolated impacts but also synergistic effects within the gating mechanism. We believe our method is significantly different from the NAEE [1] method which also evaluates the sensitivity of experts on the calibration dataset. The aforementioned comparisons highlight the significant novelty of our method.
> > 2. **Our novelty is more reflected in the unique contributions in terms of methodology and theory.** We propose two novel methods—early truncation and Router-Guided Importance Sampling (RGIS)—to further improve the accuracy and efficiency of Monte Carlo sampling based Shapley value estimation. Early truncation leverages the observation of performance collapse in low-expert coalitions and is backed by a novel error bound theorem (Theorem 1), which theoretically justifies its use and demonstrates bounded estimation error. Similarly, RGIS exploits MoE's gating probabilities as a natural prior for importance sampling via a Plackett-Luce model, with proven unbiasedness and variance reduction (Theorem 2). These are not generic heuristics; they are tailored to MoE's router-driven sparsity, reducing computational complexity while improving accuracy, as validated in our ablation studies and experiments, where Shapley-MoE outperforms baselines by significant margins.
> >
> > We hope that the further clarifications provided above can alleviate your concerns regarding the novelty of our method.
> >
> > > One of the major claims of this work is to reduce the computational overhead of finding prunable experts. That is why I asked the authors to compare the results with the router-norm-based method.
> >
> > 1. Our method can obtain a high-performance pruned MoE model with tens of billions of parameters in just tens of minutes, and the aforementioned pruning overhead remains minimal and acceptable. Although the pruning overhead of our method is not as low as that of router-norm-based method, our method can achieve higher accuracy compared to router-norm-based method. We believe that the slight increase in computational overhead is worthwhile for achieving further accuracy gains.
> > 2. Additionally, in practice, the computational costs during deployment are often much higher than those during training. For example, OpenAI spent more than 4 billion dollars this year to run the inference workload of ChatGPT, while 3 billion dollars was spent to complete the full training of ChatGPT as well as other models (source: OpenAI training and inference costs could reach \\$7bn for 2024, AI startup set to lose \\$5bn - report). Therefore, we believe that it is worthwhile to spend a slightly higher pruning cost to obtain a pruned model with higher accuracy, as it brings better deployment benefits.

---

> > ### Author Response · Authors · 2025-08-07
> > **Response to Reviewer vwFD (2/3)**
> >
> > > The authors claim that the router-norm-based method is not data-free, as it uses finetuning dataset. However, my view regarding this work is different from the authors', as here the fine-tuning dataset is used for training (which indeed requires a sufficient amount of data), not for identifying the sensitivity of the experts.
> >
> > We claim that the method proposed in [2] does not require data during the pruning process. However, the process of obtaining the fine-tuned model does require data. Therefore, the method proposed in [2] is not data-free for the entire process. We believe our view on this point is consistent with that of the reviewer.
> >
> > > The authors provided a comparison with the specified method. However, improving the results by only 0.20% with the additional complexity and computational overhead does not seem a major advantage. Also, the authors did not compare for other pruning ratios.
> >
> > 1. **The improvement is significant.** The accuracy of the unpruned model is 98.89%. When the model pruning ratio is 35%, the accuracy of the method proposed in [2] is 98.13% (with an accuracy loss of 0.76%), while the accuracy of our method is 98.32% (with an accuracy loss of 0.57%). Under the setting of a lower pruning ratio, the inherent capabilities of the model are better preserved, leaving less room for further improvement. Our method further narrows the accuracy gap between the pruned model and the original model, and we believe the above improvement is significant rather than trivial.
> > 2. **It will not introduce significant additional complexity and computational overhead.** The end-to-end time of our method when used for pruning the V-MoE model on CIFAR-10 is only 10.2 minutes (measured on a single A100 80GB GPU), which we believe is minimal computational overhead. Compared with the computational overhead of the method in [2], the slight increase in computational overhead is worthwhile for achieving further accuracy gains.
> > 3. **Comparative results of other pruning ratios.** As per your suggestion, in addition to the above-provided experimental results of V-MoE with a model pruning ratio of 35%, we further present the experimental results of V-MoE with model pruning ratios of 19% and 50% on CIFAR-100. We pruned the model without performing post-pruning fine-tuning. The experimental results are as follows:
> > |Pruning ratio|0%|19%|50%|
> > |-|-|-|-|
> > |Method in [2]|92.57%|91.87%|85.71%|
> > |**Ours**|**92.57%**|**92.12%**|**87.55%**|
> >
> >     From the experimental data in the table, we can draw the following conclusions:
> >
> >     Our method outperforms the method proposed in [2] under different pruning ratios. At a pruning ratio of 19%, it further narrows the accuracy gap with the original model, achieving a 0.25% accuracy improvement compared to the method in [2]. **Moreover, under the setting of a higher pruning ratio, the advantage of our method are further highlighted, with an accuracy improvement of 1.84% compared to the method proposed in [2].** The above results further verify the effectiveness and superiority of our method. We will include the above experimental results in the final version of our paper.

---

> ### Author Response · Authors · 2025-08-07
> **Response to Reviewer vwFD (3/3)**
>
> > Finally, I still believe that providing some sort of theoretical performance guarantee of the pruned MoE model itself while using the Shapley value would greatly benefit the work.
>
> Thank you for your valuable suggestion. According to your suggestion, we have provided the theoretical performance guarantee of the pruned MoE model itself while using the Shapley value.
>
> **Theorem [Performance Preservation Bound]**
>
> Let $\mathcal{E}$ denote the complete set of experts in the MoE model, and $\mathcal{E}\_{\text{pruned}} = \mathcal{E} \setminus \\{E_i^l \mid \hat{\phi}\_{E_i^l} < \epsilon\\} (l\in\\{1,2,\ldots,L\\}, i\in\\{1,2,\ldots,n\\})$ be the pruned expert set with Shapley value threshold $\epsilon > 0$. Under the following conditions:
>
> 1. The Shapley value estimator satisfies $|\hat{\phi}\_{E_i^l} - \phi\_{E_i^l}| \leq \eta$ for all $E_i^l \in \mathcal{E}$ with probability $\geq 1 - \delta$.
> 2. The cooperative game is additive, i.e., $V(\mathcal{S}) = \sum_{E_i^l \in \mathcal{S}} \phi\_{E_i^l}$ for any $\mathcal{S} \subseteq \mathcal{E}$.
>
> Then with probability at least $1 - \delta$, the performance degradation after pruning satisfies:
> $$
> V(\mathcal{E}) - V(\mathcal{E}_{\text{pruned}}) \leq N(\epsilon + \eta)
> $$
> where $N = |\mathcal{E}|$ is the total number of experts.
>
> **Proof.**
>
> Under the additivity assumption,
> $$
> V(\mathcal{E}\_{\text{pruned}}) = \sum_{E_i^l \in \mathcal{E}\_{\text{pruned}}} \phi\_{E_i^l}.
> $$
>
> Therefore, the performance degradation is
> $$
> V(\mathcal{E}) - V(\mathcal{E}\_{\text{pruned}}) = \sum_{E_i^l \notin \mathcal{E}\_{\text{pruned}}} \phi\_{E_i^l}.
> $$
>
> For each pruned expert $E_i^l \notin \mathcal{E}\_{\text{pruned}}$ (i.e., $\hat{\phi}\_{E_i^l} < \epsilon$), we have
> $$
> \phi\_{E_i^l} \leq \hat{\phi}\_{E_i^l} + \eta < \epsilon + \eta,
> $$
> with probability at least $1 - \delta$. Let $R = |\\{E_i^l \notin \mathcal{E}\_{\text{pruned}}\\}|$ be the number of pruned experts. Then,
> $$
> \sum_{E_i^l \notin \mathcal{E}\_{\text{pruned}}} \phi\_{E_i^l} < R (\epsilon + \eta) \leq N (\epsilon + \eta).
> $$
>
> Therefore we complete the proof.
>
> We believe that the aforementioned theory has further enhanced the theoretical contribution of our work. We will incorporate the above content into the final version of our paper and would like to express our sincere gratitude again for your valuable suggestion.
>
> **We hope that the above response has addressed your concerns and look forward to your further response.**

---

> > ### Comment · Reviewer_vwFD · 2025-08-07
> >
> > Thank you for your response. Can you give some reasoning why condition 2 may hold?

---

> > > ### Author Response · Authors · 2025-08-07
> > > **Response to Reviewer vwFD**
> > >
> > > **Thank you for your further response.**
> > > 1. The efficiency property of Shapley value states: $\sum_{E_i^l \in \mathcal{E}} \phi\_{E_i^l} = V(\mathcal{E}) - V(\emptyset)$. When $V(\emptyset) = 0$ (which is reasonable since a model with no experts would have undefined or catastrophically poor performance, making $V(\emptyset) = 1/\text{PPL} \approx 0$), this simplifies to $\sum_{E_i^l \in \mathcal{E}} \phi\_{E_i^l} = V(\mathcal{E})$. The condition 2 extends this relationship from the complete set $\mathcal{E}$ to any subsets $\mathcal{S} \subseteq \mathcal{E}$.
> > > 2. The true Shapley value of a subset of experts, $V(\mathcal{S})$, is a highly complex, non-linear, and non-monotonic function. It depends on intricate synergies and redundancies between the experts in $\mathcal{S}$. For example, two experts might be highly redundant, so $V({E_1, E_2}) < \phi\_{E_1} + \phi\_{E_2}$. Conversely, two other experts might be highly synergistic, so $V({E_3, E_4}) > \phi\_{E_3} + \phi_{E\_4}$. Capturing all these interaction effects in a formal bound would be incredibly difficult, if not impossible, and the resulting formula would be too complex to offer any clear theoretical derivation. To balance and simplify such complex synergistic effects and redundant relationships, we propose the additivity assumption (i.e., condition 2).
> > > 3. The additivity assumption, $V(\mathcal{S}) = \sum_{E_i^l \in \mathcal{S}} \phi\_{E_i^l}$, replaces this complex reality with a simple linear model. This simplification allows the proof to proceed in a few straightforward algebraic steps, leading to the clean and intuitive result: $V(\mathcal{E}) - V(\mathcal{E}_{\text{pruned}}) \leq N(\epsilon + \eta)$. The core message—that the performance loss is bounded by the number of pruned experts multiplied by the sum of the Shapley value estimated error and threshold—is made transparent.
> > >
> > > **We hope that the above response has addressed your concerns and look forward to your further response.**

---

### Official Review · Reviewer_3TXS · 2025-07-02

**Clarity:** 3
**Significance:** 2
**Originality:** 3
**Rating:** 2
**Confidence:** 4

**Summary:**

The paper tackles the problem of pruning Mixture-of-Experts (MoE) language models, which often waste memory and compute on redundant experts. Instead of relying on heuristics or exhaustive search, the authors frame expert selection as a cooperative game and propose Shapley-MoE—a pruning framework that scores each expert by its Shapley value, i.e., its average marginal contribution to the model’s performance (measured as $1/\text{perplexity}$). Each expert is treated as a “player,” and the model’s performance over any subset of experts acts as the payoff function. The Shapley value formula precisely captures an expert’s average benefit across all coalitions.Exact Shapley values require evaluating $2^N$ subsets and are intractable for hundreds of experts. The authors approximate them by sampling permutations and computing marginal gains, reducing the complexity from exponential to $\mathcal{O}(MN)$.Theorem 1 quantifies how much accuracy is lost when Monte Carlo sampling stops early once a sampled expert subset degrades performance below a threshold. Theorem 2 analyzes the estimator that draws permutations according to router activation probabilities.

**Questions:**

It would be helpful if the authors could further clarify the novelty of their approach, especially in relation to prior work that employs Shapley values for pruning neurons or convolutional filters. In addition, providing some intuition or guidance on the sample complexity would make it easier to assess the method’s practicality. It would also be useful to offer theoretical or empirical justification for the assumption that activation frequency reflects an expert’s true contribution. Moreover, it remains unclear whether expert rankings based on the 1/PPL objective are indicative of broader model importance. Finally, some insight into how often early truncation occurs and the extent to which it may bias perplexity estimates would be appreciated.

**Ethical Concerns:**

["NO or VERY MINOR ethics concerns only"]

**Final Justification:**

Many thanks to the authors thorough response and clear clarifications.

I will remain my score.

**Limitations:**

yes

**Quality:**

2

**Strengths And Weaknesses:**

The paper is methodologically sound and well‑executed. The use of Shapley values is mathematically justified, and the combination of Monte Carlo estimation with early truncation and router‑guided importance sampling (RGIS) forms a coherent, scalable pipeline. The writing is clear and the empirical evaluation is thorough. However, the core idea lacks novelty. The use of Shapley values for model pruning has appeared in prior work on neurons and convolutional filters, and extending this to MoE experts is a conceptually straightforward adaptation. The main contributions—early truncation and RGIS—are useful heuristics, but they are presented as practical add‑ons rather than fundamental algorithmic innovations. The theoretical analysis also has limitations. Theorem 2 only provides a one‑sided variance bound, without quantifying the expected gains, making the practical benefit of RGIS hard to predict. Theorem 1 ignores the additional variance introduced by the importance weights in RGIS, which may lead to large hidden constants. Furthermore, there is no theoretical justification that Shapley values computed with respect to 1/PPL will transfer to downstream tasks. Finally, despite reduced cost, the method still relies on Monte Carlo estimates.

---

> ### Author Rebuttal · Authors · 2025-07-31
>
> **Thanks for your careful review and comments!**
>
> > W1 & Q1: Core idea lacks novelty. The use of Shapley values for model pruning has appeared in prior work on neurons and convolutional filters, and extending this to MoE experts is a conceptually straightforward adaptation. The main contributions—early truncation and RGIS—are useful heuristics, but they are presented as practical add‑ons rather than fundamental algorithmic innovations.
>
> We appreciate the reviewer's insightful comment regarding prior applications of Shapley value in neurons or convolutional filters pruning. The similarity between these works and ours is that both determine the importance of model components based on the Shapley value, thereby pruning the unimportant model components.
>
> **However, we respectfully argue that extending Shapley values to MoE pruning is far from a straightforward adaptation due to the unique structural and computational characteristics of MoE architectures.** Unlike dense neural networks or convolutional layers, where components (e.g., neurons or filters) are always active and contribute uniformly, MoE experts operate in a sparse, conditionally activated manner governed by gating mechanisms. This introduces interdependencies among experts that create a fundamentally different "cooperative game" compared to prior settings. **Directly applying existing Shapley-based pruning methods would overlook these dynamics, leading to inefficient or inaccurate importance estimation.** The results of our ablation experiments in Table 4 show that our method has higher accuracy and lower pruning cost compared with the original Shapley value pruning method based on Monte Carlo sampling.
>
> **Moreover, our main contributions—early truncation and Router-Guided Importance Sampling (RGIS)—are not merely practical "add-ons" but principled innovations designed to address the exponential computational challenges unique to large-scale MoE.** Early truncation leverages the observation of performance collapse in low-expert coalitions and is backed by a novel error bound theorem (Theorem 1), which theoretically justifies its use and demonstrates bounded estimation error. Similarly, RGIS exploits MoE's gating probabilities as a natural prior for importance sampling via a Plackett-Luce model, with proven unbiasedness and variance reduction (Theorem 2}). These are not generic heuristics; they are tailored to MoE's router-driven sparsity, reducing computational complexity while improving accuracy, as validated in our ablation studies and experiments, where Shapley-MoE outperforms baselines by significant margins.
>
> **In essence, while building on the foundational idea of Shapley value, our work introduces a novel framework that adapts and extends them to the distinct challenges of MoE pruning, with theoretically grounded estimate accelerations that enable scalability to MoE.** We believe this represents a meaningful advancement in efficient MoE deployment.
>
> > W2: Theoretical analysis also has limitations. Theorem 2 only provides a one‑sided variance bound, without quantifying expected gains, making practical benefit of RGIS hard to predict.
>
> Thank you for your insightful feedback on Theorem 2. However, we would like to make the following clarification.
> 1. **Expected gains of RGIS**: We agree that a more precise quantification of variance reduction would strengthen the analysis. The one-sided bound was designed to offer a rigorous guarantee: under the reasonable assumption of positive correlation between activation frequencies and true Shapley value, RGIS ensures variance is no worse than standard Monte Carlo sampling. However, we also acknowledge that this bound does not specify the magnitude of improvement, which depends on factors such as strength of correlation, specific models/datasets, etc., and this increases the difficulty of our theoretical analysis.
> 2. **Practical benefits of RGIS**: To address above, our ablation studies in Table 4 provide empirical quantification of RGIS's benefits: RGIS improves the accuracy of our Shapley-MoE method. Removing RGIS would lead to an increase of 1.08 in perplexity. This means that RGIS can accelerate Monte Carlo sampling, as Monte Carlo sampling requires more sampling steps to achieve the same level of accuracy.
>
> > W3: Theorem 1 ignores additional variance introduced by importance weights in RGIS, which may lead to large hidden constants.
>
> We appreciate reviewer's astute observation regarding potential impact of importance weights on variance in Theorem 1, which indeed focuses on early truncation mechanism and does not explicitly account for RGIS-induced variance, which may lead to large hidden constants. However, we would like to make the following clarification.
>
> 1. **Clarify the theorem's scope**: Theorem 1 specifically bounds the estimation error for early truncation in standard Monte Carlo sampling, without incorporating RGIS. Its purpose is to justify truncation mechanism's efficiency and bounded error independently.
> 2. **Theoretical mitigation of variance**: Theorem 2 demonstrates that RGIS reduces variance compared to standard Monte Carlo under premise of assuming a complete sampling process, which mitigates additional variance introduced by importance weights in RGIS as well as magnitude of hidden constants.
> 3. **Empirical validation**: Table 4 provides empirical validation for early truncation and RGIS. When both early truncation and RGIS are removed, perplexity increases by 3.49. Our ablation studies further demonstrate that RGIS with truncation yields accurate Shapley estimates and better pruning results than naive Monte Carlo, suggesting combined approach is robust empirically even if hidden constants are not fully minimized.
>
> > W4: There is no theoretical justification that Shapley values computed with respect to 1/PPL will transfer to downstream tasks.
>
> While Shapley values in our method are computed using V=1/PPL (an intrinsic measure of language modeling proficiency), we justify its transferability via both theory and empirics:
> 1. **Theoretically**, perplexity is a standard, intrinsic evaluation metric for LLMs, directly measuring a model's ability to predict held-out data and reflecting its overall language modeling quality. By using V=1/PPL, we quantify expert contributions to the model's core generative capabilities, which form the foundation for downstream performance. This choice aligns with prior work in LLM pruning and quantization, where perplexity is commonly used as proxies for compressed LLM performance.
> 2. **Empirically**, we evaluate zero-shot performance on seven downstream tasks and knowledge reasoning, arithmetic, and code generation capabilities in experimental section. experiments show that the Shapley value calculated based on 1/PPL reflects the expert's contribution well, and the MoE model with a higher Shapley value has higher accuracy on downstream datasets.
>
> > W5: Despite reduced cost, the method still relies on Monte Carlo estimates.
>
> The exact computation of Shapley values has time complexity $\mathcal{O}(2^N)$, which is infeasible for MoE where $N$ typically exceeds $10^3$. Monte Carlo sampling approximates it, reducing complexity to $\mathcal{O}(MN)$, which $M$ is the number of samples. Thus, Monte Carlo estimation is essential to significantly lower computational costs. In addition, in our response to Weakness 1 from Reviewer ovkS, we also demonstrated that error between Shapley value estimated using Monte Carlo sampling and ground-truth Shapley value is very small.
>
> > Q2: Providing some intuition or guidance on sample complexity would make it easier to assess the method’s practicality.
>
> The Monte Carlo-based Shapley value estimation has time complexity $\mathcal{O}(MN)$, where $M$ is the sample number and $N$ is the total number of experts in MoE. Figure 2 shows pruning costs under different $M$ settings; increasing $M$ raises costs. Thus, balancing search efficiency and model performance, we set $M=20$. Our method's end-to-end time is only 36 minutes, indicating good practicality.
>
> > Q3: It would also be useful to offer theoretical or empirical justification for assumption that activation frequency reflects an expert’s true contribution.
>
> The relationship between activation frequency and expert contribution is justified as follows:
> 1. **Theoretically**, the gating network in MoE routes inputs to the most appropriate experts, prioritizing those that contribute most to performance. Frequently activated experts are those the model deems essential for handling inputs. Therefore, the above theoretical evidence supports the assumption that activation frequency reflects an expert’s true contribution.
> 2. **Empirically**, we present extensive results of pruning experts in MoE models based on activation frequency in Section 4. Frequency-based pruning outperforms random-based pruning, which empirically validates that activation frequency can indeed reflect an expert’s true contribution.
>
> > Q4: It remains unclear whether expert rankings based on 1/PPL objective are indicative of broader model importance.
>
> Extensive experiments demonstrate that expert rankings based on 1/PPL and Shapley value effectively reflect importance in broader MoE models. This includes results of our method on popular open-source MoEs (Qwen, DeepSeek and Mixtral series) in Sections 4.2 and 4.3, and multimodal MoE in Section I.1.
>
> > Q5: Some insight into how often early truncation occurs and the extent to which it may bias perplexity estimates would be appreciated.
>
> Early truncation is applied during each Monte Carlo sampling when catastrophic performance collapse occurs. Table 4 presents ablation results: removing it increases pruning cost (time rises from 36 to 62 minutes) and degrades pruned MoE performance (perplexity increases by 1.67).
>
> **We hope that our response has addressed your concerns, and we kindly ask for a more favorable evaluation of our work.**

---

> ### Comment · Reviewer_3TXS · 2025-08-05
>
> Many thanks to the authors for your thorough response and clear clarifications.

---

> > ### Author Response · Authors · 2025-08-07
> > **Looking forward to your response**
> >
> > Thank you for your efforts in reviewing our paper. We have provided responses to your concerns. We would apreciate it if you could let us know whether our replies have addressed your questions. Please feel free to raise any further issues.

---

### Official Review · Reviewer_ovkS · 2025-07-05

**Clarity:** 4
**Significance:** 3
**Originality:** 3
**Rating:** 4
**Confidence:** 2

**Summary:**

This paper proposes a new pruning method for Mixture-of-Experts models to discover more important experts. The authors introduce the Shapley value to estimate the importance of each expert and approximate the Shapley value by using a Monte Carlo sampling strategy. Early Truncation and Router-Guided Importance Sampling are also developed for the improvement of sampling accuracy and efficiency. Theoretical and experimental analyses demonstrate that the proposed method showcases a promising way for MoE model pruning.

**Questions:**

- Please provide more experimental results for different percentages of pruning.
- Please illustrate the motivation for using Monte Carlo sampling to approximate the Shapley estimation
- Can this method be employed in other LLM architectures? If yes, how to achieve this goal?

**Ethical Concerns:**

["NO or VERY MINOR ethics concerns only"]

**Final Justification:**

After reading the rebuttal, I think the authors clearly explain their contributions and address some of my concerns, especially the comparison of the ground-truth Shapley value and the estimated value. The motivation of using Monte Carlo sampling presented in the main paper would be helpful. The question 3 I asked is about the application of other LLM architectures with no expert, which may be presented as potential limitations. I'm maintaining my overall score and I hope these suggestions are helpful for improving clarity and presentation.

**Limitations:**

Yes

**Quality:**

3

**Strengths And Weaknesses:**

Strengths
- The paper innovatively extends Shapley value to evaluate the importance of experts and proposes a Monte Carlo sampling approach for approximating Shapley value, which incorporates early truncation and router-guided importance sampling.
- The authors provide theoretical guarantees for the error bound of early truncation Shapley estimation as well as the unbiasedness and variance reduction of RGIS.
- The experimental results illustrate the robustness and the effectiveness of the proposed method.
Weaknesses
- The differences between the ground-truth and estimated value: The authors use Monte Carlo sampling to approximate the Shapley value, and calculating the ground-truth Shapley value is an NP-hard problem. Despite this, can the authors measure the differences between the ground-truth Shapley value and the estimated value?
- More experimental results: The experiments are conducted in the setting of 25% and 50% pruning, what about 75% pruning or else? Can you provide a performance trend in terms of different percentages of pruning?

---

> ### Author Rebuttal · Authors · 2025-07-31
>
> **We appreciate the time and effort you put in reviewing our work, as well as your detailed comments and valuable questions. We understand the concerns you raised and we're pleased to address these concerns.**
>
> > Weakness 1: The differences between the ground-truth and estimated value: The authors use Monte Carlo sampling to approximate the Shapley value, and calculating the ground-truth Shapley value is an NP-hard problem. Despite this, can the authors measure the differences between the ground-truth Shapley value and the estimated value?
>
> Thank you for your suggestions. We have conducted experimental measurements below. Considering that the time complexity of calculating the ground-truth Shapley value using Equation. 3 is $\mathcal{O}(2^N)$, where $N$ is the total number of experts in the MoE model, and the number of experts in each MoE layer of the Qwen series and DeepSeek series MoE models exceeds 60. Therefore, we used Equation. 3 in our paper to calculate the ground-truth Shapley value of the 8 experts in the first MoE layer of the Mixtral-8x7B model, and adopted our proposed Shapley-MoE method to calculate the estimated Shapley value. The experimental results are as follows:
> |Expert id|0|1|2|3|4|5|6|7|
> |-|-|-|-|-|-|-|-|-|
> |Ground-truth|2.54e-4|3.03e-4|-2.94e-4|-1.19e-5|2.68e-3|7.24e-5|8.08e-5|-6.54e-5|
> |Estimated|2.66e-4|3.12e-4|-2.74e-4|-1.45e-5|2.70e-3|7.67e-5|8.00e-5|-6.14e-5|
> |Difference|+0.12e-4|+0.09e-4|+0.20e-4|-0.26e-5|+0.02e-3|+0.43e-5|-0.08e-5|+0.40e-5|
>
> From the above experimental results, we can draw the following conclusions:
> 1. **The difference between the ground-truth Shapley value and the estimated value is very small, which proves that using our Shapley-MoE method to estimate the Shapley value is highly effective.** In addition, since calculating the ground-truth Shapley value is an NP-hard problem, it is not feasible in practice. However, our method provides an effective reference scheme for estimating Shapley value in practice.
> 2. Both the ground-truth and estimated Shapley values follow the descending order of 2/4/3/5/6/7/1/0. **This means that the experts of MoE pruned based on the estimated Shapley value are the same as those pruned based on the ground-truth Shapley value.** In this case, the accuracy of the pruned model obtained by Shapley-MoE is the same as that of the pruned model based on the ground-truth Shapley value. This proves that the Shapley value estimated by our Shapley-MoE method will not affect the final pruning accuracy.
>
> Thank you again for your insightful opinions. We will incorporate the above experimental results into the final version of our paper.
>
> > Weakness 2 & Question 1: More experimental results: The experiments are conducted in the setting of 25% and 50% pruning, what about 75% pruning or else? Can you provide a performance trend in terms of different percentages of pruning?
>
> Thank you for your suggestion. **We have shown the experimental results under different pruning rates in Section 4.3. In Table 3, we show the WikiText-2 perplexity performance of the pruned Mixtral-8x7B model under a wider range of pruning rates, from 12.5% to 75%.** The experimental results show that:
> 1. As the pruning rate increases, the perplexity of the pruned model shows an upward trend, which indicates that the performance of the pruned model shows a downward trend with the increase of the pruning rate.
> 2. Across different pruning rate settings, the perplexity of the pruned models obtained by our Shapley-MoE method consistently remains lower than that of pruned models obtained by other methods, which fully demonstrates the robustness and effectiveness of our Shapley-MoE method.
>
> > Question 2: Please illustrate the motivation for using Monte Carlo sampling to approximate the Shapley estimation.
>
> Thank you for your question. **The motivation for using Monte Carlo sampling to approximate the Shapley estimation is to make the calculation of Shapley value feasible for large-scale MoE models.** While the Shapley value theoretically provides a metric for quantifying expert contributions, exact computation through Eq. 3 requires evaluating all $2^N$ expert subsets, which is a prohibitive proposition for modern MoE architectures where total expert count $N$ often exceeds $10^3$. This exponential complexity motivates the development of efficient approximation methods. Therefore, we employ the Monte Carlo sampling approach to estimate Shapley value through empirical expectation over expert permutations. By averaging over $M$ permutations, Monte Carlo sampling reduces the computational complexity from $\mathcal{O}(2^N)$ to $\mathcal{O}(MN)$, making the calculation of Shapley value feasible for large-scale MoE models.
>
> > Question 3: Can this method be employed in other LLM architectures? If yes, how to achieve this goal?
>
> Thank you for your question.
> 1. Our method is applicable to expert pruning for various MoE model architectures. **In Sections 4.2 and 4.3, we present the experimental results of our method on various popular open-source MoE architectures,** including the Qwen series (Qwen1.5-MoE-A2.7B, Qwen2-57B-A14B, and Qwen3-30B-A3B), the DeepSeek series (DeepSeekMoE-16B and DeepSeek-V2-Lite), and the Mixtral series (Mixtral-8x7B). **In addition, our method is also applicable to expert pruning for multimodal MoE models.** In Section I.1, we present the experimental results of the MoE-LLaVA-Qwen-1.8B-4e model.
> 2. Our method is applicable to expert pruning for all models with mixture-of-experts architectures that include experts. However, for models with other dense LLM architectures, such as the LLaMA-1/2 series, our method is not applicable because these models do not contain experts.
>
> **Finally, we hope that our response has addressed your concerns, and we kindly ask for a more favorable evaluation of our work. We will incorporate the above clarification into the final version of our paper. Thank you!**

---

### Official Review · Reviewer_ysJ4 · 2025-07-17

**Clarity:** 3
**Significance:** 2
**Originality:** 3
**Rating:** 4
**Confidence:** 4

**Summary:**

This paper proposes Shapley-MoE, a pruning framework that quantifies each expert’s importance using Shapley values. The authors estimate each expert’s marginal contribution to model performance via Monte Carlo approximation, and incorporate two key optimizations, Early Truncation and Router-Guided Importance Sampling, to reduce the theoretical exponential complexity to an approximate quadratic level while preserving accuracy. Experiments are conducted on various public MoE models including Qwen, DeepSeek, and Mixtral, demonstrating improvements in perplexity and zero-shot task accuracy under 25%–75% pruning rates, as well as reductions in GPU memory usage and inference latency.

**Questions:**

1. How do expert pruning, channel pruning, quantization, and their combinations compare in terms of memory-performance and speed-performance trade-off curves?
2. How does Shapley-MoE perform against baseline methods (e.g., Random, Frequency) under random datasets and task distributions, where its calibration strategy may not align with task-specific characteristics?

**Ethical Concerns:**

["NO or VERY MINOR ethics concerns only"]

**Final Justification:**

The authors’ response has addressed my concerns well, and I will maintain my score.

**Limitations:**

Please refer to the weaknesses above.

**Paper Formatting Concerns:**

No major formatting issues noticed.

**Quality:**

3

**Strengths And Weaknesses:**

Strengths
- Applying Shapley values to assess expert importance in MoE is a novel and well-motivated approach, showing promising results in both efficiency and performance.
- The paper includes comprehensive experiments and ablation studies to support its conclusions.

Weaknesses
- The paper lacks a thorough discussion on the necessity of expert pruning. Given that the core motivation is to reduce memory usage and improve inference speed, it would be more convincing to include comparisons with alternative approaches such as channel pruning and quantization. These methods may achieve similar resource reductions with potentially smaller performance degradation. Although the authors suggest that Shapley-MoE can be combined with such methods, it remains unclear whether expert pruning lies on the Pareto frontier when considering the trade-off between performance and efficiency.
- The paper acknowledges that different tasks may require different calibration datasets for expert pruning, which aligns with the conceptual understanding of MoE experts. However, this raises questions about the fairness of comparisons with baseline methods like Random or Frequency-based pruning. Specifically, under randomly selected datasets and tasks, does Shapley-MoE still demonstrate consistent advantages over prior approaches?

---

> ### Author Rebuttal · Authors · 2025-07-31
>
> **We greatly appreciate your time, detailed comments, and valuable suggestions. We understand the concerns you raised and we're pleased to address these concerns.**
>
> > Weakness & Question 1: The paper lacks a thorough discussion on the necessity of expert pruning. Given that the core motivation is to reduce memory usage and improve inference speed, it would be more convincing to include comparisons with alternative approaches such as channel pruning and quantization. These methods may achieve similar resource reductions with potentially smaller performance degradation. Although the authors suggest that Shapley-MoE can be combined with such methods, it remains unclear whether expert pruning lies on the Pareto frontier when considering the trade-off between performance and efficiency. How do expert pruning, channel pruning, quantization, and their combinations compare in terms of memory-performance and speed-performance trade-off curves?
>
> Thank you for your thoughtful and constructive feedback. According to your suggestions, we demonstrate the compression effects of expert pruning, channel pruning, and quantization on the Qwen2-57B-A14B model. Specifically, we pruned 25% of the experts using our method, pruned 25% of the channels in each expert using magnitude pruning, and performed 4-bit quantization for the Qwen2-57B-A14B model using the GPTQ method. The experimental results are as follows:
> |Method|WikiText-2 perplexity|Average zero-shot accuracy|Memory (GB)|Speedup|
> |-|-|-|-|-|
> |Qwen2-57B-A14B|5.86|65.39%|110.29|1.00$\times$|
> |GPTQ quantization|6.14|65.01%|30.38|1.27$\times$|
> |Channel pruning|7.99|61.40%|90.14|1.21$\times$|
> |Shapley-MoE|6.78|63.05%|88.46|1.26$\times$|
> |Shapley-MoE w/GPTQ quantization|7.12|62.78%|25.37|1.51$\times$|
>
> From the data in the table, we can observe the following:
>
> (1) **Our method has advantages over channel pruning:** Our Shapley-MoE method outperforms the channel pruning method, with a perplexity that is 1.21 lower and an average zero-shot accuracy that is 1.65% higher. We believe that although channel pruning operates at a finer-grained level to prune the model, it disrupts important experts, leading to a more severe performance degradation. In contrast, our method preserves the full capabilities of relatively important experts, which enables it to achieve better performance.
>
> (2) **Compared with the original model, our method has only a slight performance drop, while it can significantly reduce the memory usage and improve the inference speed:** Compared with the original model, our method only increases the perplexity by 0.92 and decreases the average zero-shot accuracy by merely 2.34%. Meanwhile, it can achieve a 19.79% reduction in memory usage and a 1.26$\times$ improvement in inference speed. Compared with quantization, the aforementioned significant reduction and inference speed improvement can be obtained on any device without the need for a specific inference engine.
>
> (3) **Our method still has advantages over quantization:** The perplexity of our method is only 0.64 higher than that of the quantization method, and the average zero-shot accuracy is only 1.96% lower. Furthermore, combining our method with the quantization method can further advance the Pareto frontier of MoE model compression. Shapley-MoE with AWQ quantization only increases the perplexity by 1.26 and decreases the average zero-shot accuracy by 2.61% compared with the original model, while achieving a 77% reduction in memory usage and a 1.51$\times$ improvement in inference speed. This further demonstrates the superiority of our method.
>
> Furthermore, we would also like to make the following clarifications regarding the necessity of expert pruning and its comparison with channel pruning and quantization methods.
> 1. **On the necessity of expert pruning**: We agree that a more explicit discussion of why expert pruning is essential for MoE would enhance the paper. MoE architectures, as noted in our methodology section, feature a large number of sparsely activated experts (often exceeding 1000 in models like DeepSeekMoE-16B or Qwen2-57B-A14B), leading to significant memory and computational overhead during inference. Unlike dense models, MoE's conditional activation means many experts contribute marginally or redundantly, making expert-level pruning a natural and efficient way to reduce parameters while preserving the model's sparse computation benefits. This is particularly crucial for deployment on resource-constrained devices.
> 2. **Comparison with pruning and quantization method**: You are correct that methods like channel pruning (e.g., removing less important channels in FFNs) and quantization (e.g., 4-bit or 8-bit weight compression) can achieve similar resource reductions. However, these approaches are orthogonal to expert pruning: channel pruning operates at a finer granularity within experts, potentially leading to higher performance degradation in sparse MoE settings due to disruption of expert specialization, while quantization reduces bit-width but specific quantized kernels are required to achieve model memory reduction and improved inference speed. In contrast, expert pruning is hardware-agnostic and can reduce model memory usage and enhance inference speed on any device.
>
> Thank you again for your constructive suggestions. We will include the above results in the final version of our paper.
>
> > Weakness & Question 2: The paper acknowledges that different tasks may require different calibration datasets for expert pruning, which aligns with the conceptual understanding of MoE experts. However, this raises questions about the fairness of comparisons with baseline methods like Random or Frequency-based pruning. Specifically, under randomly selected datasets and tasks, does Shapley-MoE still demonstrate consistent advantages over prior approaches? How does Shapley-MoE perform against baseline methods (e.g., Random, Frequency) under random datasets and task distributions, where its calibration strategy may not align with task-specific characteristics?
>
> We appreciate the reviewer’s insightful comment regarding the role of calibration datasets in expert pruning. According to your suggestions, we verify the effectiveness of our method on random datasets, we randomly selected total 128 examples from C4, Pile, and PTB to form the calibration dataset. These datasets are also commonly used as calibration datasets in previous compression methods (such as Wanda [3], SparseGPT [4], and OmniQuant [5]). We pruned 25% of the experts in Qwen1.5-MoE-A2.7B model. The experimental results are as follows:
> |Method|WikiText-2 perplexity|Average zero-shot accuracy|
> |-|-|-|
> |Random|11.01|54.79|
> |Gating Score|10.51|55.12|
> |Frequency|10.17|55.51|
> |Expert Trim|10.41|55.01|
> |**Shapley-MoE**|**9.80**|**57.03**|
>
> **Our Shapley-MoE method outperforms other baseline methods.** For example, compared with the best Frequency baseline, Shapley-MoE has a perplexity that is 0.37 lower and an average zero-shot accuracy that is 1.52% higher. We believe Shapley-MoE maintains advantages due to its game-theoretic foundation: it quantifies experts' marginal contributions to overall model performance, capturing inter-expert collaborations that random or frequency-based methods overlook. Frequency-based approach only consider isolated activation counts and gating score-based approach only consider isolated gating weight magnitude, which may not reflect true importance in diverse coalitions, while random pruning ignores any structure and task. Therefore, our method has advantages in any data distribution.
>
> Furthermore, we would also like to make the following clarifications regarding the use of the calibration dataset in our paper.
> 1. **To ensure fairness in our comparisons, all baseline methods (including Random and Frequency-based pruning) were evaluated using the same calibration dataset as Shapley-MoE. We follow the settings of Expert pruning methods for MoE models (such as NAEE [1] and CD-MoE [2] method), randomly sampling 128 examples from the C4 training dataset as calibration data.**
>
> 2. In addition, the calibration datasets used in our paper are all datasets frequently employed in previous compression works. C4 dataset has been widely utilized in methods such as Wanda [3], SparseGPT [4] and OmniQuant [5].
>
> 3. Furthermore, NAEE [1] found that using the MATH dataset for calibration during the expert pruning process of MoE models can improve the accuracy of the pruned models on mathematical tasks, as this enables the identification of experts that are more important for mathematical tasks. Our paper follows this setup and employs domain-specific calibration datasets to improve the accuracy of the pruned model on domain-specific tasks.
>
> [1] Not All Experts are Equal: Efficient Expert Pruning and Skipping for Mixture-of-Experts Large Language Models. ACL 2024.
>
> [2] Condense, don’t just prune: Enhancing efficiency and performance in moe layer pruning. arXiv.
>
> [3] A Simple and Effective Pruning Approach for Large Language Models. ICLR 2024.
>
> [4] SparseGPT: Massive Language Models Can be Accurately Pruned in One-Shot. ICML 2023.
>
> [5] Omniquant: Omnidirectionally calibrated quantization for large language models. ICLR 2024.
>
> **Finally, we hope that our response has addressed your concerns, and we kindly ask for a more favorable evaluation of our work. We will incorporate the above clarification into the final version of our paper. Thank you!**

---

> ### Author Response · Authors · 2025-08-07
> **Last two day reminder**
>
> Dear Reviewer ysJ4,
>
> We have put in many efforts to address the raised questions and concerns. As the NeurIPS rebuttal period is approaching its end, we kindly remind you to review our submitted response. Your feedback is essential for finalizing our work.
>
> Thank you for your attention.
>
> Best regards,
>
> Submission20 Authors

---

### Decision · Program_Chairs · 2025-09-17

**Decision:**

Accept (poster)

**Comment:**

This paper proposes Shapley-MoE, an efficient pruning method for MoE models inspired by cooperative game theory. The authors estimate each expert’s marginal contribution to model performance via Monte Carlo approximation, and incorporate two key optimizations, Early Truncation and Router-Guided Importance Sampling, to reduce the theoretical exponential complexity to an approximate quadratic level while preserving accuracy. Experiments on Qwen, DeepSeek, and Mixtral demonstrate its effectiveness.

Stengths:
1. novel methods with clear motivations
2. comprehensive evaluation results across different model families and datasets

Weaknesses:
1. Some reviewers pointed out that some empirical/theoretic analyses should be carried out for better understanding the internal working mechanism of the proposed method.